

# Assessing stratospheric transport in the CMAM30 simulations using ACE-FTS measurements

Felicia Kolonjari[1], David A. Plummer[2], Kaley A. Walker[1], Chris D. Boone[3], James W. Elkins[4], Michaela I. Hegglin[5], Gloria L. Manney[6,7], Fred L. Moore[8,9], Diane Pendlebury[10], Eric A. Ray[8,9], Karen H. Rosenlof[8], and Gabriele P. Stiller[11]

[1]Department of Physics, University of Toronto, Toronto, Canada
[2]Climate Research Division, Environment and Climate Change Canada, Victoria, Canada
[3]Department of Chemistry, University of Waterloo, Waterloo, Canada
[4]Global Monitoring Division, NOAA Earth System Research Laboratory, Boulder, USA
[5]Department of Meteorology, University of Reading, Reading, UK
[6]NorthWest Research Associates, Socorro, USA
[7]Department of Physics, New Mexico Institute of Mining and Technology, Socorro, New Mexico, USA
[8]Chemical Sciences Division, NOAA Earth System Research Laboratory, Boulder, USA
[9]Cooperative Institute for Research in Environmental Sciences, University of Colorado Boulder, Boulder, USA
[10]Air Quality Research Division, Environment and Climate Change Canada, Toronto, Canada
[11]Institute of Meteorology and Climate Research, Karlsruhe Institute of Technology, Karlsruhe, Germany

*Correspondence to:* Kaley A. Walker (kaley.walker@utoronto.ca)

**Abstract.**

Stratospheric transport in global circulation models and chemistry-climate models is an important component in simulating the recovery of the ozone layer as well as changes in the climate system. The Brewer-Dobson circulation is not well constrained by observations and further investigation is required to resolve uncertainties related to the mechanisms driving the circulation.

5   This study has assessed the specified dynamics mode of the Canadian Middle Atmosphere Model (CMAM30) by comparing to the Atmospheric Chemistry Experiment Fourier Transform Spectrometer (ACE-FTS) profile measurements of CFC-11 ($CCl_3F$), CFC-12 ($CCl_2F_2$), and $N_2O$. In the CMAM30 specified dynamics simulation, the meteorological fields are nudged using the ERA-Interim Reanalysis and a specified tracer was used for each species, with hemispherically-defined surface measurements used as the boundary condition. A comprehensive sampling technique along the line-of-sight of the ACE-FTS

10   measurements has been employed to allow for direct comparisons between the simulated and measured tracer concentrations. The model consistently overpredicts tracer concentrations in the lower stratosphere, particularly in the Northern Hemisphere winter and spring seasons. The three mixing barriers investigated, including the polar vortex, the extratropical tropopause, and the tropical pipe, show that there are significant inconsistencies between the measurements and the simulations. In particular, the CMAM30 simulation exhibits too little isentropic mixing in the June-July-August season.



## 1 Introduction

Interest in stratospheric transport has increased over the last 20 years as a result of significant developments in stratosphere-resolving general circulation models (GCMs) and chemistry-climate models (CCMs) (e.g., Pawson et al., 2000; Eyring et al., 2005; SPARC-CCMVal, 2010; Gerber, 2012). Projections of stratospheric ozone and climate rely on the ability of these models

to simulate stratospheric transport and chemistry. The distribution of trace gases in the stratosphere is primarily controlled by the Brewer-Dobson Circulation (BDC), which is generally characterized by tropospheric air entering the stratosphere in the tropics, poleward transport, and descent in the midlatitude and polar regions of the winter hemisphere (e.g., Plumb, 2002; Butchart, 2014, and references therein). The BDC describes the primary features of the stratospheric circulation, based upon a conceptual model proposed to explain observations of ozone and water vapour (Dobson et al., 1929; Brewer, 1949; Dobson,

1956). Over the last decade, BDC modeling studies have reached a consensus regarding how the stratospheric circulation will respond to changes in anthropogenic climate forcing (Butchart, 2014). It is now understood that the residual circulation as well as quasi-isentropic mixing are key factors to understanding the structure of the BDC (McLandress et al., 2011; Butchart, 2014; Abalos et al., 2015; Ploeger et al., 2015a, b; Oberländer-Hayn et al., 2016). Plumb (2002) and Shepherd (2007) state that two-way mixing between the tropics and extratropics and the rapid stirring of air parcels are important components of stratospheric

transport. The influence of planetary waves on stirring is predominately in the winter midlatitude surf zone (McIntyre and Palmer, 1983, 1984) but synoptic-scale wave activity occurs throughout the year in the subtropical lower stratosphere and its influence can extend upwards to 25 km (Haynes and Shuckburgh, 2000). It has been argued by Shepherd and McLandress (2011) that the greenhouse gas induced warming of the climate system has led to an upward displacement of the critical layers for wave breaking. Subsequently, it has been suggested that the BDC changes are characterized more by a vertical lifting of

rather than an acceleration of the meridional circulation (Oberländer-Hayn et al., 2016).

Plumb (2002) and Birner and Bönisch (2011) identified two distinct pathways within the BDC. These are the "deep branch", defined as the poleward transport in the winter hemisphere extending into the middle and upper stratosphere, and the "shallow branch(es)", defined as multiple pathways of faster poleward transport that are observed in both hemispheres throughout the year and are generally restricted to the lower-to-middle stratosphere. It is likely that the shallow branches are driven by

Rossby-wave pumping on a synoptic scale (Plumb, 2002; Butchart, 2014). As part of the Climate Chemistry Model Validation (CCMVal) project, Lin and Fu (2013) investigated simulated changes in the BDC by considering three branches separately; the transition, shallow, and deep branches. They found that changes in the transition and shallow branches of the BDC were consistent with the increase of greenhouse gas concentrations and the trends were associated with changes in subtropical jets and tropical upper tropospheric temperatures, which is also consistent with the mechanism described by Shepherd and McLan-

dress (2011). The acceleration of the deep branch is consistent with that of the transition and shallow branches but is seasonally modulated by changes in ozone concentrations with the exact mechanisms yet to be determined (Lin and Fu, 2013). The mechanisms that lead to an acceleration or deceleration in the deep branch remain unresolved (Shepherd and McLandress, 2011; Lin and Fu, 2013). Observational evidence seems to indicate a deceleration in the deep branch (Engel et al., 2009; Hegglin et al., 2014). However, there is a fair degree of confidence that the acceleration of the shallow branch is likely since it is driven





by the vertical lifting mechanism proposed by Shepherd and McLandress (2011) that is related to large-scale changes and also has been diagnosed from changes in stratospheric constituent distributions (Hegglin et al., 2014).

Despite significant progress in modeling, the BDC has been poorly constrained by observations (Butchart, 2014). A direct comparison to determine how the BDC and quasi-horizontal mixing combine to produce the distribution of long-lived tracers with tropospheric sources is not possible. Therefore, a number of observational techniques have been used to investigate stratospheric transport characteristics, such as age of air diagnostics (Stiller et al., 2008; Engel et al., 2009), tropical lower stratosphere ascent rates (Mote et al., 1996; Niwano et al., 2003), and descent rates in the Antarctic polar vortex (Abrams et al., 1996b; Allen et al., 2000; Kawamoto and Shiotani, 2000), and in the Arctic polar vortex (Abrams et al., 1996a; Greenblatt et al., 2002; Ray et al., 2002; Greenblatt, 2003). Except for Stiller et al. (2008), these observations do not provide global seasonally-resolved quantitative estimates of the BDC (Butchart, 2014). Thus, it has proven difficult to deduce changes in the strength of the BDC using available measurements (e.g., Engel et al., 2009; Diallo et al., 2012; Seviour et al., 2012; Stiller et al., 2012; Haenel et al., 2015).

Fundamental questions remain as to the mechanisms driving the stratospheric circulation because there has been evidence of changes in the BDC that has not been projected by models (e.g., Butchart, 2014; Mahieu et al., 2014). It is clear that the transport of chemical tracers will be impacted by changes in the BDC, which will in turn influence ozone recovery projections, lifetimes of ozone depleting gases, and mass exchange between the troposphere and stratosphere (Butchart, 2014). Understanding how the structure of the BDC will change depends greatly upon the ability to simulate its current behaviour. This is typically assessed by investigating how capable a model is at simulating long-lived tracers; in particular, assessing the characteristics of the simulated tracers such as concentrations and behaviour at the mixing barriers in the stratosphere (i.e. the polar vortex edge, the extratropical tropopause, and the tropical pipe).

In this study, measurements of the long-lived chlorofluorocarbons, $CCl_3F$ (CFC-11) and $CCl_2F_2$ (CFC-12), and $N_2O$ from the Atmospheric Chemistry Experiment Fourier Transform Spectrometer (ACE-FTS) are used to evaluate the specified dynamics simulation mode of the Canadian Middle Atmosphere Model (CMAM30). Using these global measurements, areas in which simulated tracers agree with observations and where improvements are needed have been investigated. Since ACE-FTS measurements are vertically resolved, these data are useful for testing model simulations (e.g., Hegglin and Shepherd, 2007; Manney et al., 2009; Strahan et al., 2011); however, care must be taken in the methods used for these comparisons. In addition to comparison methods, external tools are useful to the interpretation of differences between CMAM30 and ACE-FTS. An idealized stratospheric model, the Tropical Leaky Pipe (TLP) model, described by Ray et al. (2016) has been used to test factors contributing to the differences observed between ACE-FTS and CMAM30.

Many limb-viewing satellite missions, including ACE, are contributing data to the SPARC Data Initiative, whose purpose is to compile and assess a repository of climatologies for comparison with model output (Hegglin et al., 2013; Tegtmeier et al., 2013; Neu et al., 2014; Tegtmeier et al., 2016). Some of the instruments involved in the initiative, such as ACE-FTS, do not cover all latitudes and altitudes each month. This matter has been the subject of recent studies (Toohey and von Clarmann, 2013; Toohey et al., 2013; Millán et al., 2016). They have shown that the impacts of sampling patterns of various instruments




must be considered when performing comparisons because the model simulates data points that are evenly spaced throughout each latitude range in the zonal mean at each pressure level while the measurements can represent a subset of that domain.

This work addresses issues related to climatological comparisons in several ways: first, by using a nudged version of the CMAM; second, by breaking out the species of interest from the halocarbon family arrangement in the model; and third, by
sampling the model output along the individual measurement profile pathways through the atmosphere. The latter addresses sampling issues identified by Toohey and von Clarmann (2013), Toohey et al. (2013), and Millán et al. (2016). By isolating the model output in this way, the transport and chemical processes in CMAM can be evaluated. This type of assessment of CMAM simulations has not been possible until recently because of the typically free-running nature of the model simulations. The CMAM specified dynamics simulation has been investigated in a few recent studies: McLandress et al. (2014) evaluated the
polar cap mesospheric transport and midlatitude mean zonal winds, and long term observational records of water vapour and ozone were also used to evaluate the CMAM30 run by Hegglin et al. (2014) and Shepherd et al. (2014), respectively. Additionally, Pendlebury et al. (2015) investigated the CMAM30 polar regions using satellite data, including ACE-FTS. Comparisons of the CMAM30 simulations to observations remain limited in their extent, a gap in which this work attempts to fill.

This paper is structured as follows: Section 2 describes the tools used in this study including measurements from ACE-FTS,
CMAM30 simulations, and the TLP model simulations. Section 3 examines methods of sampling and comparison techniques and considers the impact of the sampling of ACE-FTS. Section 4 examines the measured and simulated zonally-averaged morphologies. Section 5 investigates the three barriers to mixing: the polar vortex, the extratropical tropopause, and the tropical pipe. In section 6, the TLP model is used to investigate changes in tropical upwelling and quasi-isentropic mixing required to allow the model to more effectively simulate the lower stratosphere. Finally, the results are summarized and discussed in
Section 7.

## 2 Tools

### 2.1 ACE-FTS

The ACE mission on-board the Canadian satellite SCISAT was launched to investigate the distribution of upper tropospheric and stratospheric ozone with the goal to further our understanding of the chemical and dynamical processes that influence its
behaviour (Bernath, 2017). ACE entered a circular low-earth orbit (650 km, 74° inclination), on 12 August 2003, to observe the Earth's atmosphere using the solar occultation technique (Bernath et al., 2005). The high resolution Fourier transform spectrometer, ACE-FTS, is the primary instrument on SCISAT. It has a high spectral resolution (0.02 cm$^{-1}$) and a spectral range of 2.2-13.3 $\mu$m (750-4400 cm$^{-1}$) (Bernath et al., 2005). ACE-FTS does not require filters for its operation, which allows it to measure solar absorption spectra for dozens of atmospheric constituents simultaneously. ACE-FTS is ideal for studying
the vertical structure of constituent gases from cloud tops to 100 km; it is particularly useful in the upper troposphere and lower stratosphere because of its high vertical resolution (Hegglin et al., 2008). The ACE-FTS products are derived from the solar absorption spectra measured and include the vertical profiles of temperature, pressure, and the concentration expressed as a volume-mixing ratio (VMR) for several dozen molecules of atmospheric interest over latitudes from 85° N to 85° S (Bernath



et al., 2005). These data products are useful for the study of climatologies (e.g Allen et al., 2009; Jones et al., 2012; Koo et al., 2017), trends (e.g., Brown et al., 2011), and lifetimes (e.g., Brown et al., 2013), among other applications (e.g., Hegglin and Shepherd, 2007; Hegglin et al., 2009; Brown et al., 2014; Hoffmann et al., 2014; Hegglin et al., 2014).

The version 3.0 ACE-FTS retrievals of CFC-11 ($CCl_3F$), CFC-12 ($CCl_2F_2$), and $N_2O$ based on spectra recorded between June 2004 and May 2010 have been used throughout this work. A description of the retrieval process used for ACE-FTS is provided by Boone et al. (2005) and Boone et al. (2013). The earlier work details the retrieval process for version 2.2 of the data while the latter describes improvements that have been implemented for the recent versions. The CFC-11 retrieval ranges from 5 km to a maximum of 28 km in the tropical latitudes but is limited in the stratosphere at higher latitudes due to low concentrations. CFC-12 is retrieved between 5 km and 36 km, while the $N_2O$ retrieval covers 5 km to 95 km. Due to the

vertical limitation of the CFC-11 retrieval, this work focuses on the upper troposphere and lower stratosphere (5 km to 30 km). The validity of these measurements has been investigated in several studies. Mahieu et al. (2008) compared both the ACE-FTS CFC-11 and CFC-12 v2.2 products to the FIRS-2 and MkIV Interferometer measurements. They found ACE-FTS to be approximately 10% lower than FIRS-2 between 12 km and 16 km in the case of CFC-11. Using a non-coincident technique, Velazco et al. (2011) found agreement to be better than 20% between ACE-FTS and MkIV over the range of 17 km to 24 km.

The CFC-12 comparisons made by Velazco et al. (2011) and Mahieu et al. (2008) show a consistent difference with ACE-FTS approximately 10% lower than MkIV. Using a climatological validation approach within the SPARC Data Initiative to compare the CFC-11 and CFC-12 products of HIRDLS, MIPAS, and ACE-FTS, Tegtmeier et al. (2016) found excellent agreement in the lower stratosphere (up to 50 hPa) and increasing positive deviations above this level to around 20% from the multi-instrument mean. Strong et al. (2008) provided an extensive validation of the ACE-FTS v2.2 $N_2O$ product using satellite, aircraft, balloon

and ground-based FTIR measurements. Differences observed were typically within +15% (Strong et al., 2008). The work of Velazco et al. (2011) was consistent with the results of Strong et al. (2008). Waymark et al. (2013) compared the previously validated CFC-11 and CFC-12 v2.2 products with the v3.0 products used in this work and found that around 15 km there is a slight increase in CFC-11 in the new product, bringing it closer to the correlative measurements used in the studies described above. Similarly for CFC-12, Waymark et al. (2013) found an increase of 2-5% between 6 km and 22 km. They also found an

approximately 10% decrease in the concentration of $N_2O$ above 35 km in v3.0, bringing the differences found in Strong et al. (2008) to within approximately +5%.

A set of quality control flags for the ACE-FTS products on the 1-km retrieval grid are available (Sheese et al., 2015). The version 1.1 flags were applied to the data used in this work by removing profiles that contained a flag between 4 and 7, as recommended by Sheese et al. (2015). The method rejects a maximum of 6% of data over all the species retrieved. For CFC-11,

CFC-12, and $N_2O$, 1.7%, 2.1%, and 4.3% of the data are rejected (Sheese et al., 2015). In addition to the quality control flags, derived meteorological products from GEOS 5.2.0, based on the techniques described in Manney et al. (2007), are used here along with the geographic location information to account for the geographic extent and meteorological context of ACE-FTS profiles.



## 2.2 Canadian Middle Atmosphere Model

### 2.2.1 The model

CMAM is a freely-running CCM based on an upwardly-extended version of the Canadian Centre for Climate Modelling and Analysis (CCCma) third-generation Atmospheric General Circulation Model (Beagley et al., 1997; Scinocca et al., 2008). The

chemistry includes the $O_x$, $HO_x$, $NO_x$, $ClO_x$, and $BrO_x$ catalytic cycles that control ozone in the stratosphere; the chemistry of $N_2O$, $CH_4$, and seven long-lived halocarbon species; and a representation of heterogeneous chemistry on background stratospheric sulphate aerosols and on polar stratospheric clouds (deGrandpré et al., 2000; Jonsson et al., 2004). CMAM has been used extensively to investigate the middle atmosphere and to study complex processes of the climate system (e.g., Austin et al., 2003; Vyushin et al., 2007; Plummer et al., 2010; McLandress et al., 2011). Results from the CMAM have also been assessed

during two phases of the CCMVal project and, more recently, the Chemistry-Climate Model Initiative (CCMI). The extensive investigation of dynamical and chemical processes in CCMs that took place during CCMVal-2 are detailed in the SPARC report (SPARC-CCMVal, 2010). For the simulations used here, CMAM was run at a T47 spectral resolution, equivalent to approximately 3.75° x 3.75°, with 71 vertical levels topping out at 0.08 Pa (approximately equivalent to 95 km in altitude).

  Due to the chaotic nature of the atmospheric circulation, free running models are unable to reproduce the day-to-day evolu-

tion of the atmosphere. Therefore, simulated fields, such as tracer concentrations, cannot be compared to observations directly or on a day-to-day basis. There has recently been an effort to circumvent this limitation by constraining the evolution of the circulation and temperatures in CCMs with fields from reanalysis data sets through the use of Newtonian relaxation (e.g., McLandress et al., 2014; Shepherd et al., 2014), known colloquially as 'nudging'. The ability to constrain the dynamical fields to follow the observations more closely enables direct model-measurement comparisons of chemical tracers in the model by

eliminating the internal variability in the simulated circulation. This type of specified dynamics simulation has been used here to allow for space and time matched comparisons of output from CMAM with ACE-FTS observations over the June 2004 to May 2010 period.

### 2.2.2 The specified dynamics simulation

The specified dynamics version of the CMAM, which is referred to here as CMAM30, uses meteorological fields from the

ERA-Interim reanalysis (Dee et al., 2011) to constrain the dynamical fields while the chemical fields are allowed to freely evolve. The model horizontal winds and temperature are nudged towards the ERA-Interim fields. The six-hourly reanalysis data is linearly interpolated in time to produce fields for nudging at intermediate time steps. The relaxation is applied at only the synoptic scales and larger by constraining only wavenumbers up to 21. The application of nudging on only the large scales and the use of a relaxation time constant of 24-hours has been found to produce root-mean-square differences between the

CMAM30 and the reanalysis comparable to those found between different reanalysis datasets for fields such as temperature and vorticity (Merrifield et al., 2012). In addition, some minor adjustments were made to the global average temperature in the ERA-Interim fields at 5 hPa and above to remove discontinuities associated with changes in the observing system as described





by McLandress et al. (2014). Sea surface temperatures and sea-ice were specified using the HadISST dataset (Rayner et al., 2003).

The standard CMAM chemical mechanism uses a lumping approach for the halocarbon tracers to reduce the number of chemical species that must be transported by the model. A limited number of halocarbons are explicitly treated by the chemical scheme and the remaining long-lived halocarbons are combined into the model species based on their 'fractional release values' (Schauffler et al., 2003). The concentration specified as a lower boundary condition is increased so that the total amount of organic chlorine (or bromine) of all halocarbons represented by the model species is conserved. For example, the model explicitly treats the chemistry of CFC-12 ($CCl_2F_2$), but the concentration of CFC-12 was increased to account for the additional chlorine carried by CFC-113. Because of the time-varying contribution of the individual halocarbons to the tropospheric concentration of the model species, it is not possible to re-scale the concentration of the model species to recover a concentration that could be compared with observations. Therefore, to directly compare the model halocarbon concentrations with observations, a parallel set of halocarbons was added to the model that explicitly represents individual halocarbon species. These parallel species undergo the appropriate chemical reactions using the photolysis rates and concentrations calculated by the full model chemical mechanism, though the reactions of the parallel species do not feed into the concentration of other model species. The CMAM30 simulation including the additional explicit halocarbon species will be referred to as CMAM30HR for the remainder of this work.

### 2.2.3 Influence of the surface boundary conditions

In CMAM, global average concentrations are typically applied as the lower boundary condition for long-lived species, such as $N_2O$. To capture the inter-hemispheric differences in tropospheric concentrations of the halocarbon tracers, the parallel species have separate Northern and Southern Hemisphere surface mixing ratios imposed as lower boundary conditions. The lower boundary conditions were derived from the annual average hemispheric mixing ratios from the National Oceanographic and Atmospheric Administration's HAlocarbon and other Trace Species (HATS) program (Elkins et al., 1993; Montzka et al., 1996). The annual average values were linearly interpolated in time to calculate an instantaneous surface-layer mixing ratio for the model. The model mixing ratio in the lowest six model layers, approximately the lowest 1 km, was relaxed towards the specified concentration with a time constant that increased from 25 days near the equator to 12 hours at 25 degrees of latitude. In CMAM30HR, all important losses for the species of interest have been considered. The photolysis rates and reaction rates have been updated to the values from JPL-2010 (Sander et al., 2011). The chemical losses of CFC-11 are dominated by reactions with $O(^1D)$ and photolysis in the mid-to-lower stratosphere, particularly in the tropical region. The chemical losses of CFC-12 and $N_2O$ are similar to that of CFC-11, except they generally occur higher in the stratosphere. CFC-11, CFC-12, and $N_2O$ losses are insignificant in the troposphere.

To ensure the efficacy of the boundary conditions applied to the CMAM30HR simulation, the model output was compared to measurements at surface monitoring sites using data from the NOAA HATS program. The monthly mean measurements of $N_2O$, CFC-11, and CFC-12 have been compared to the monthly mean in the CMAM30HR output between 2004 and 2010. Because of the way the surface boundary condition was imposed, each site was compared to the lowest model level of the





closest grid point in the CMAM30HR output. The relative differences have been calculated by subtracting the measurement from the simulation and dividing by the mean of the two for each month in the time series. The differences over time were compared and no trend in the differences was observed. The comparisons of each trace gas at each site are summarized in Fig. 1, ordered by latitudinal location. Data included here are averaged over the 2004 to 2010 period and error bars (±) indicate one

standard deviation.

Both CFC-11 and CFC-12 simulations have reasonably small differences, generally less than 1%, from the surface measurements. Over the time period compared, CMAM30HR appears to overpredict CFC-11 at all HATS sites while the CFC-12 comparisons are not significantly different from zero for all but two sites in the Southern Hemisphere. The $N_2O$ comparisons show that the model consistently underpredicts the concentrations at 11 of the 13 sites. There appears to be some latitudinal

dependence in the comparisons of $N_2O$ which may be caused by the application of a globally-averaged boundary condition in the run. Generally the differences for all the species shown are within ±1%, which is to be expected since the model boundary conditions were derived from these measurements.

### 2.2.4    Influence of nudging on the age of stratospheric air

The mean age of stratospheric air, that is the average time elapsed since the last time an air parcel was in the troposphere, can

provide a diagnostic for determining differences in isentropic transport and mixing between different model runs (Hall and Plumb, 1994; Waugh and Hall, 2002). The stratospheric age of air in the model is derived from an idealized $SF_6$ tracer whose lower boundary condition linearly increases over time. In this section, the CMAM30, rather than CMAM30HR, mean age of air is used since no transport changes were made between the two runs of the model. Averaged between 2004 and 2010, Fig. 2a is the zonal distribution of the CMAM30 mean age subtracted from the mean age from an identical, but freely-running,

version of the CMAM using the same specified sea surface temperatures and sea-ice data. The differences in age range from approximately −1.25 year to +1 years, where the positive differences indicate areas where the air in CMAM is older than that in CMAM30, and the negative differences indicate areas in which the CMAM30 age is older than in the CMAM age. In general, the nudging of CMAM appears to affect the Southern Hemisphere more than the Northern Hemisphere. Below 50 hPa in the tropics and midlatitude regions, the difference between the age in the two versions of the model is close to zero. Above

50 hPa in the tropics and midlatitudes and above 150 hPa in the polar regions, air in CMAM is older than that in CMAM30, with peaks occurring around the surf zones in the stratosphere. This implies that for the majority of the lower stratosphere the process of nudging leads to an apparent decrease in the CMAM age of air. The cause of this has not been fully explained in the literature at this time; however, it can be speculated that nudging the model to the reanalysis could be a source of artificial drag that drives the BDC to be more rapid than in the free-running version of the model.

In the polar regions of Fig. 2a, the comparisons exhibit a different behaviour in the lowermost stratosphere relative to the rest of the stratosphere. The differences are close to zero at the lowest pressure level shown (400 hPa). Above approximately 300 hPa, the CMAM30 air is older than the CMAM air and younger above approximately 150 hPa. These differences, while strongest at the latitudes poleward of 60°, can extend to approximately 40° latitude in both hemispheres. Figures 2b and 2c show the monthly evolution at 80°S and 80°N, respectively. Below 150 hPa, the CMAM30 air appears to be older than the air



in the free run and this tends to be pronounced during the respective summer months in each hemisphere. At 80°S, the pattern of differences in the age of air change in altitude over time. At approximately 150 hPa in May, the air in CMAM30 is older than the air in CMAM. This difference appears at approximately 100 hPa by October, with a larger magnitude. In November and December, during Austral spring, the age difference remains such that the CMAM30 air is older than the air in CMAM.

In Fig. 2c, the evolution of the differences in age of air at 80°N appears restricted to the same altitudes but the seasonal timing of the pattern is similar. The difference peaks in spring/summer and dissipates through the fall and winter. The prevalence of the older air in the polar lowermost stratosphere in the nudged run is significant because, throughout the stratosphere, air in CMAM30 is younger than that in CMAM. It is known that the freely running CMAM has a cold bias inside the Antarctic vortex. McLandress et al. (2012) suggest that there may be missing gravity wave drag (GWD) in the Southern Hemisphere

based on comparisons of the free running model simulations and reanalysis data. By adding this missing GWD through the nudging to reanalysis data, downwelling between 70°S to 90°S is increased, leading to higher temperatures – a reduction in the cold bias – during September and October. The increased downwelling pushes the older air deeper into the lowermost stratosphere, causing the observed differences in age between the two versions of CMAM.

In general, synoptic-scale waves are filtered out close to the tropopause and only planetary-scale waves can propagate further

up into the stratosphere (Dickinson, 1969). Plumb (2002) showed that synoptic-scale wave drag drives the lower branch of the BDC, while the drag that drives the deeper parts of the BDC are associated with planetary wave drag. CMAM30 appears to reproduce the upper troposphere simulated in CMAM and, to some extent, the lower branch of the BDC as well. This is evidenced by the near-zero differences in Fig. 2a between 50°S and 50°N and between 100 hPa and 50 hPa; however, the absolute ages in this region tend to be quite small. Understanding the impact of the nudging on the age of air provides the

basis for an interpretation of the isentropic transport and mixing differences between the two model runs. While it is difficult to quantify the extent to which the differences in age of air would change tracer concentrations at a given location, it is necessary to consider these results when considering implications for the free-running model, particularly for the deep branch of the BDC. Based on Fig. 2, CMAM30 clearly has older air in the extra-tropical lowermost stratosphere. It is potentially caused by either stronger downwelling of the older air from above, consistent with a stronger BDC, or increased isentropic mixing of

tropospheric air from lower latitudes (e.g. Hegglin and Shepherd, 2007). Therefore, the differences in age appear to suggest a slower shallow branch or a faster deep branch of the BDC.

### 2.3 Tropical Leaky Pipe Model

A modified TLP model (Ray et al., 2016) is used to interpret the differences between the CMAM30HR simulations and the ACE-FTS measurements. The modified TLP is based on a set of three coupled one dimensional equations relating transport

between the tropics and each hemispheric extratropical region (Plumb, 1996; Neu and Plumb, 1999; Hall and Waugh, 2000). The model includes advection, vertical diffusion, and horizontal mixing between the extratropics and the tropics. Significant changes to the modified version of the TLP model include common pressure coordinates in all regions and the addition of particle trajectories with photochemistry. The modification was done to allow for direct comparisons between TLP output and other models and/or measurements. The Lagrangian approach is described in detail by Ray et al. (2016). The tropical





boundaries in the TLP model averages were chosen based on observational estimates of the upwelling region (Ray et al., 2016). The model was run with a vertical resolution of 200 m and a maximum altitude of 40 km above tropopause; however, the results included here are limited to 30 km in altitude above the tropopause. To ensure the effectiveness of the TLP as an interpretation tool, Ray et al. (2016) established that the TLP could accurately simulate the CMAM30HR output with its mean

circulation and a TLP-derived mixing parameter as an input. The mixing parameter was derived from a suite of simulations conducted with the TLP at varying amounts of mixing. The resultant best match to the averaged 2004-2010 CMAM30HR CFC-11, CFC-12, and age of air profiles was the level of mixing selected.

Diagnosing the causes of discrepancies between measurements and model output requires a complete separation of the effects of the strength of the BDC and the mixing. A simplified model, such as the TLP, is useful to interpret differences

between measurements and CCMs because of the complexity of wave activity contributing to stratospheric mean circulation and mixing. It would not be prudent to adjust model parametrizations in CMAM to modify wave breaking because many aspects of the model climatology would be impacted with no way of separating the effects (Ray et al., 2016). Therefore, a suite of model runs were computed with the TLP to test a range of mean circulation strengths and mixing efficiencies. The TLP runs began with the CMAM30HR best fit to the TLP model and then the best combination was selected to match the ACE-FTS

measurement profiles of CFC-11 and CFC-12, as well as an age-of-air estimate derived from balloon-measurements. In section 6, the TLP simulations are used to investigate the behaviour of the tropical pipe in CMAM30.

## 3    Comparison methods and sampling considerations

### 3.1    Measurement-model comparison techniques

Two of the comparison techniques used in this study are described here; the first is the comparison of zonal means, and the

second is the computation of joint probability density functions.

### 3.1.1    Zonal mean comparison technique

To assess the transport and chemistry in CMAM30HR, measurements of $N_2O$, CFC-12, and CFC-11 are compared with simulated concentrations in latitude-pressure coordinates. A common method of visualizing the distribution of long-lived trace gases is the zonal mean cross section. In this work, data from the ACE-FTS profiles, sampled CMAM30HR profiles, and

the relative difference profiles were averaged in 5° latitude bins and over 18 pressure levels (equally distributed in the log of pressure from 450 hPa to 10 hPa), corresponding to altitude ranges from approximately 5 km to 30 km. In these plots, color contours indicate the VMR of the species. The comparison of the ACE-FTS measurements and the subsampled CMAM30HR output is shown as the average of the differences, defined as CMAM30HR minus ACE-FTS divided by the mean of the model output and measurements. The altitude of the average thermal tropopauses is typically indicated by a blue line. All

measurements and subsampled model output between June 2004 and May 2010 have been included, representing an average of six years of observations and simulations.





### 3.1.2 Joint probability density functions

Tracer-tracer correlations have been used in a number of studies to identify transport and mixing characteristics in the stratosphere and to derive climatologies from sparse data (e.g., Plumb and Ko, 1992; Plumb, 1996; Toon et al., 1999; Sankey and Shepherd, 2003; Hegglin and Shepherd, 2007). Plumb and Ko (1992) showed that long-lived species exhibit compact corre-

lations even with varying meteorological conditions, minimizing discrepancies resulting from sampling and daily variations; thus, sparse measurements, such as those from aircraft, can be useful in model assessment studies (e.g., Sankey and Shepherd, 2003). The correlations used here are $N_2O$/CFC-11 and include only the stratospheric data available (data located at altitudes above 2 km above the thermal tropopause). The correlations produced from both the ACE-FTS measurements and CMAM30HR simulations exhibit compact relationships that tend to be densely populated. Determining whether the model can

capture the clustering in addition to the overall shape is important to understanding whether the stratosphere is well-reproduced by the model. Understanding the density distribution of a data set is particularly useful for tracer-tracer relationships with compact correlations. Following the methods of Sparling (2000) and Hegglin and Shepherd (2007), normalized joint probability distribution functions (JPDFs) have been calculated for the ACE-FTS and CMAM30HR correlations described above. JPDFs are two-dimensional histograms that reveal the clustering of data (Hegglin and Shepherd, 2007) and can be used to test how

well a model captures the behaviour of trace gases in the stratosphere.

### 3.2 The influence of beta angle

ACE-FTS records a series of spectra along a slanted path line-of-sight during each occultation. The length of this slanted path is different for each occultation. Each ACE-FTS spectrum is assigned a latitude and longitude at the 30-km tangent point, geometrically calculated (Boone et al., 2005, 2013). A sample year of the geometric 30-km tangent point latitudes is

provided in Fig. 3 (black circles), showing the annual repeating latitudinal coverage. The beta angle parameter, a measure of the angle between the solar vector and the satellite orbit plane, has also been included in Fig. 3 (red circles). The beta angle is an important parameter to consider because as it changes, so does the geographic distance between each spectrum acquired through the profile of any given occultation. The distance is greatest at high beta angles (both positive and negative), which occur when ACE is in view of the Sun for longer periods. Since the FTS instrument measurement frequency is held at a constant

two second interval, more measurements per profile and longer ground-paths of the retrieved profile occur at high beta angles.

    Considering the impact of observation sampling is a critical step when comparing measurements with model output. The work of Toohey and von Clarmann (2013), Toohey et al. (2013), and Millán et al. (2016) illustrate the necessity for considering the sampling patterns resulting from different measurement techniques and satellite orbits. The ground path length of a profile is considered because a single profile can be representative of more than one geographic region, typically varying more over

latitude than longitude. A refraction model is used to determine the geographic locations along the slant path of the ACE-FTS profiles (Boone et al., 2005, 2013). At the 30-km tangent altitude, it has been found that for 98% of the ACE-FTS occultations, the difference between the geometric latitude and the refraction calculation is less than $0.2°$. A useful marker of a nominal occultation length is at a beta angle of $53°$, corresponding to an occultation duration of three minutes. Occultations longer





than three minutes, at beta angles larger than 53°, measure across large spatial distances and represent approximately 12% of
the ACE-FTS data used in this work. Both horizontal and vertical variations within the CMAM30HR output will impact the
comparison to ACE-FTS measurements. The CMAM30HR fields are output on a grid with a spatial resolution of approximately
400 km. While most ACE-FTS occultations have a shorter horizontal extent than the CMAM30HR grid point footprints, there

are some occultations that fall outside of a single grid point range in the upper troposphere and lower stratosphere. For example,
between 5 km and 30 km, 15% of occultations extend across more than one CMAM30HR grid point footprint, of which 82%
are at beta angles greater than 53° and 80% are at latitudes poleward of 30°. Various model sampling techniques have been
investigated since occultations span multiple grid point footprints in both latitude and longitude, as well as vertically.

### 3.3   Comparison of sampling techniques

To determine the impact of sampling the model output at varying levels of detail, three methods were tested and compared to
the full model output (the CHAM30HR output at all latitudes and longitudes for each 5° latitude bin) between June 2004
and May 2010. All three methods began with identifying the temporally coincident three-dimensional CMAM30HR output
(latitude, longitude, pressure) for each ACE-FTS profile; the output within 3 hours of the occultation was selected with no
temporal interpolation. The three-dimensional output was interpolated in the vertical dimension to the ACE-FTS profile pres-

sure grid, which is different for each occultation since the retrievals are performed on an altitude scale. The 'basic' sampling
method involved selecting a vertical column based on the nearest neighbour grid point to the 30 km tangent point location with
no interpolation and no consideration of the vertical extent of the profile. The 'intermediate' level of sampling extracted the
vertical column based on a bilinear interpolation of the four closest grid points to the 30 km geometric tangent point but with
no consideration of the variation in geographical location of the tangent points; therefore, there was horizontal interpolation

but no vertical interpolation. The 'advanced' sampling method improves on the intermediate level of sampling by performing
the bilinear interpolation at each level of the ACE-FTS profile using the distinct geographic locations, derived from the re-
fraction model (Boone et al., 2005, 2013), for the respective level. Therefore, at each vertical level in the ACE-FTS profile, a
spatial bilinear interpolation including the four geographically closest grid points was computed to determine the comparable
CMAM30HR VMR. To illustrate the sampling effect between 450 hPa and 10 hPa (5-30 km), the relative differences in the

zonal mean of $N_2O$ over the observation period (June 2004 – May 2010) are compared in Fig. 4. The advanced method is
compared to the full output of the model (Fig. 4a), the basic model sampling (Fig. 4b) and the intermediate sampling (Fig. 4c).

The comparison of the advanced sampling and full model output in the stratosphere is dominated by the influence of the polar
vortex in both hemispheres. Generally, there is good agreement throughout the troposphere, with less than 5% differences. For
the long-lived tracers investigated in this work, the free troposphere is well mixed such that there is minimal influence of the

ACE sampling pattern. In the stratosphere, however, there are pronounced differences on the order of 20%. Air in the polar
vortex is typically representative of older air brought down from higher altitudes. Therefore, tracer concentrations within the
vortex and vortex edge tend to be significantly different from those in the mid-latitude surf zone during the winter in each
hemisphere. The differences seen in Fig. 4a occur because comparing the full output to measurements does not account for
the variability of the vortex edge in both longitude and latitude. At the edge of the vortex, tracer concentration gradients are



strong, so comparing measurements to the full output of the model will tend to smear the influence of the vortex on tracer concentrations. The differences are not symmetric latitudinally due to different dynamical conditions in each hemisphere. For example, in the Antarctic stratosphere in September there is a strong decrease in the geographic extent of the polar vortex with height such that the vortex is much wider geographically at 100 hPa than at 10 hPa. A similar phenomenon occurs in the Arctic
but it is much more variable both spatially and vertically.

The advanced sampling technique is compared to the basic sampling in Fig. 4b. The distinction between Fig. 4a and 4b is that rather than using the full model output, the nearest grid point is selected based on the geographic location of the 30 km tangent point of the ACE-FTS measurements. Even this basic level of sampling improves the comparison in the stratosphere substantially, bringing the range of differences down to $\pm10\%$. It is worth noting that the stratospheric differences in the
midlatitudes are on the same order of magnitude with a similar latitudinal pattern but of opposite sign. The differences of 5-10% are primarily negative in the Southern Hemisphere stratosphere and positive in the Northern Hemisphere. This pattern occurs because each ACE-FTS profile is tilted such that the top of the profile is always further north than the bottom of the profile, leading to a directional bias. In the Northern Hemisphere, profiles tend to 'point' toward the North Pole, therefore measurements in this hemisphere are subject to a poleward bias. In the Southern Hemisphere, profiles point toward the equator,
leading to an equatorward bias in sampling. The choice of 'closest' grid point likely biases the comparisons, leading to the differences in Fig. 4b.

Figure 5 illustrates the average latitudinal extent of occultations, between the 5 km and 30 km tangent altitudes, showing the two directionalities for each 5° latitude bin included in Fig. 4b, with error bars indicating one standard deviation from the mean latitudinal extent. A poleward bias implies that the 30 km tangent point is located poleward of the 5 km tangent point
and an equatorward bias reflects when the 30 km tangent point is located equatorward of the 5 km tangent point.

In the midlatitude region of the Northern Hemisphere, the average latitudinal extent of occultations exhibits a primarily poleward bias while the occultations in the Southern Hemisphere midlatitudes exhibit a primarily equatorward bias. The Northern Hemisphere poleward bias in Fig. 5 corresponds to the positive differences in the Northern Hemisphere midlatitude stratosphere in Fig. 4b and the Southern Hemisphere equatorward bias corresponds to the negative differences in the Southern Hemisphere
midlatitude stratosphere. Since the basic sampling is limited to the 30 km tangent altitude, the comparison to the advanced technique in Fig. 4b reflects the influence of the geographical extent of the ACE-FTS profiles. In the Northern Hemisphere, contributions from sampling the model at lower latitudes (which tend to have a higher concentration) lead to positive differences between the two sampling techniques; while in the Southern Hemisphere, contributions from sampling the model at higher latitudes (lower concentrations) lead to negative differences between the advanced and basic sampling techniques. This
nuance in the sampling pattern highlights the importance of considering the sampling pattern of the ACE-FTS occultations when comparing measurements to model output.

Figure 4c compares the advanced and the intermediate sampling. With approximately $\pm2\%$ differences between the two techniques, it is clear that the intermediate sampling technique can account for much of the geographic extent of the ACE-FTS profiles at this model resolution. However, if comparing to a model with a finer resolution or if a larger vertical extent is
considered, accounting for the full geographic extent of the profile will become more important. The more detail that is included





in the sampling of the model, the more comparable the output is to the observations. The advanced method of sampling provides the most appropriate model profiles for direct comparison between ACE-FTS and CMAM30HR. Therefore, all comparison results shown in this study utilize CMAM30HR output that has been sampled using the advanced technique.

## 4 Zonally-averaged tracer morphologies

Most long-lived tracers with tropospheric sources exhibit quantitatively similar behaviour in the upper troposphere and lower stratosphere. In the context of a zonally-averaged tracer morphology, the equator-to-pole gradients of tracer isopleths that are created by the diabatic circulation in the stratosphere are readily observed. By choosing to sample the CMAM30HR output as described above, the behaviour of $N_2O$, CFC-12, and CFC-11 can be investigated thoroughly.

### 4.1 General features of tracer morphology comparisons

The zonally-averaged distribution of $N_2O$ is presented in Fig. 6. The $N_2O$ simulated by CMAM30HR is shown in Fig. 6a, the ACE-FTS measurements are shown in Fig 6b, and the average of the profile differences within each $5°$ latitude bin is shown in Fig. 6c. Both the ACE-FTS measurements and the CMAM30HR distribution of $N_2O$ in Fig. 6 show many of the features that are expected of a long-lived tracer with a tropospheric source and chemical losses that occur primarily in the stratosphere. The distributions show a decrease in concentration of $N_2O$ with altitude at all latitudes, and also moving from the equator poleward

at each pressure level and in each hemisphere. There is a hemispheric asymmetry in the decrease with altitude beyond the tropical region. The southern extratropical and Antarctic concentrations of $N_2O$ tend to decrease with altitude more rapidly in the Southern Hemisphere than in the Northern Hemisphere. This is likely caused by significant differences in the conditions of the influence of downwelling within the polar vortex between the two hemispheres. By visual comparison, the lowest concentrations observed and simulated appear to be in the Antarctic region between 30 hPa and 10 hPa and the Arctic strato-

sphere above 20 hPa. The quantitative comparison between the ACE-FTS and CMAM30HR zonal mean $N_2O$ distributions in the bottom panel of Fig. 6 reveals significant differences throughout the lower stratosphere, with the largest differences in the northern polar region. CMAM30HR simulates larger concentrations of $N_2O$ in the lower stratosphere. Upwelling in the tropics, descent in the extratropics, and mixing in the surf zone define the transport controls on the distributions in the stratosphere. The differences observed in Fig. 6c are influenced by the combined effects of these features on the measured and the simulated

concentrations of $N_2O$. Therefore, if there were no issues in the simulated stratospheric transport, the differences would be of similar magnitude to the upper troposphere comparisons (less than $\pm 5\%$), unless there was a significant flaw in the chemical losses in the model.

Investigating measurement-model comparisons using more than one trace gas leverages the varying lifetimes of and chemical processes of each gas. The comparisons of CMAM30HR and ACE-FTS, equivalent to the bottom panel of Fig. 6, are shown

in Fig. 7a-c for $N_2O$, CFC-12, and CFC-11, respectively. Each of the panels shows the differences as a percentage, but note that the scale for Fig. 7c is different. This is because the CFC-11 relative differences become large in the stratosphere where concentrations tend toward zero. All three species show good measurement-model agreement (within approximately 5%) in





the well mixed troposphere. In the tropics, the VMRs of these three species remain relatively constant up into the lower stratosphere where chemical loss processes begin to break down the compounds. Above 70 hPa in the tropics, the CFC-12 and $N_2O$ comparisons show similar agreement (on the order of 5%). However, above 50 hPa in the tropics, CFC-11 exhibits both positive and negative differences between the measurements and model simulations. These differences in CFC-11 are also observed outside the tropics above 70 hPa and are much higher (on the order of 50%). In the Northern Hemisphere extratropics, the differences are primarily positive but become more variable closer to the northern polar region. Very small concentrations above 70 hPa, which occur because of the significant photolytic losses in the tropical lower stratosphere, lead to the large magnitude of the differences in CFC-11. The irregular pattern in the CFC-11 differences is driven by the variability in the measurements as ACE-FTS reaches its detection limit.

## 4.2 Seasonality of the tracer morphology comparisons

The structure and intensity of the BDC varies seasonally. In general, the BDC is strongest in the Northern Hemisphere winter due to wave driven enhancements initiated by topography (e.g., Rosenlof, 1995; Plumb and Eluszkiewicz, 1999). It is well known that tropical upwelling is stronger in the summer hemisphere; therefore during the December-January-February (DJF) season, the upwelling is strongest in the Southern Hemisphere (e.g., Yulaeva et al., 1994). Investigating the comparisons between the CMAM30HR simulations and the ACE-FTS observations in a seasonal context helps to determine whether the differences observed earlier are related to the behaviour of the BDC. If the differences observed in Fig. 7 are driven by the simulation of the BDC in the model, it would be expected that the morphology of the seasonal differences would appear to follow the behaviour of the BDC.

For each season, Fig. 8 identifies the differences between the simulation and the measurements. The seasonal composites shown here do not fully represent the seasons because of the sampling pattern of the ACE-FTS (recall Fig. 3). However, the comparisons are relevant since the CMAM30HR output has been subsampled, as previously described. The most obvious features across all seasons in Fig. 8 are the same as those of the differences shown in Fig. 7a. There is good agreement in the lower stratosphere at all latitudes and in the tropics up to about 50 hPa. In the mid-stratosphere, CMAM30HR simulates higher concentrations of $N_2O$ than those measured by ACE-FTS. Some of the largest differences occur at the high northern latitudes during boreal winter and spring, presumably in the region of downwelling within the polar vortex.

The large disagreements in the north polar region during winter and spring indicate that the upwelling portion of the BDC across the different seasons is not well characterized by CMAM30HR. However, the shifting of the agreement in the tropical region through the seasons indicates a robust simulation. For example, the difference in the southern tropical latitudes appears small (close to 0%) up to 50 hPa and to approximately 40°S in DJF, but the agreement diminishes in this region in the March-April-May (MAM) season, presumably when the tropical upwelling begins to decline in strength and shift toward the equator. A similar pattern is observed in the Northern Hemisphere during Austral winter where the differences in the northern tropical latitudes appear to be small up to 50 hPa and to approximately 40° N in the June-July-August (JJA) and September-October-November (SON) seasons. These results support the understanding that the most rapid tropical upwelling is occurs in the summer hemisphere as first reported by Yulaeva et al. (1994).





The polar regions of each hemisphere in the comparisons of Fig. 8 are significantly different. There are negative differences at high southern latitudes in MAM, JJA, and part of SON. The differences seem quite asymmetric when compared with the results for the Northern Hemisphere. The negative differences at high southern latitudes appear to descend between MAM and JJA and begin to weaken in SON and the vortex break-down. There is also some asymmetry in the differences between 30 and

10 hPa between the Northern and Southern Hemispheres, particular in winter for each hemisphere. These differences are likely due to the behaviour of the polar vortex in each hemisphere.

Since the model run compared here has been nudged to the ERA-Interim meteorology, it cannot be simply concluded that the differences are due to the variable nature of the vortex. The vertical migration of the negative differences in the southern polar region across MAM, JJA, and SON suggests the vortex variability physical or chemical mechanism as the cause. The

negative differences mean that there is an underprediction in the CMAM30HR simulation, which could happen if air that is too old is brought down into the vortex. As the vortex forms in fall, the negative differences appear and descend through the winter, reaching a maximum latitudinal extent. The appearance of the negative differences in the comparison of $N_2O$ between the observations and the model is conspicuous because elsewhere $N_2O$ is higher in CMAM30HR throughout the lower and middle stratosphere.

In Fig. 8b and Fig. 8d, the large positive differences in the Northern Hemisphere stratosphere may be caused by too much topographic wave driving in CMAM30HR (and CMAM30). This would lead to the movement of air from the tropical region into the extratropics and polar regions, too quickly and thereby simulating higher than expected concentrations of $N_2O$ in the Northern Hemisphere stratosphere.

## 5 Comparison of mixing barriers

It is well understood that quasi-horizontal mixing flattens tracer isopleths in mixing regions and sharpens gradients at mixing barriers (e.g., Plumb, 2002). However, it can be very difficult to separate the effects of mixing barriers from the residual circulation when looking at zonal mean comparisons between measurements and models. Therefore, it is necessary to scrutinize mixing barriers individually.

### 5.1 The polar vortex

Consideration of the behaviour of the polar vortex in both hemispheres is necessary as they have atmospheric processes affect their behaviour differently over time. For this purpose, the monthly mean differences between ACE-FTS and CMAM30HR over the time period of the study have been determined for the stratospheric abundances of $N_2O$ and CFC-12. CFC-11 has been excluded from this comparison because the limited vertical extent of the sensitivity of the measurement results in too few data in the stratosphere. All comparisons shown here are profiles located poleward of $60°$, and show the mean of the difference

between the ACE-FTS and CMAM30HR profiles, expressed as a percent of the average of the two profiles. The comparisons here extend the work of Pendlebury et al. (2015), who discussed the polar region simulations in CMAM30 extensively by comparing temperature, ozone, methane, and water vapour up to 0.001 hPa with a variety of satellite instruments including



ACE-FTS. All the species investigated in Pendlebury et al. (2015) have much shorter lifetimes than those of $N_2O$ and CFC-12. The advantage of using species with long lifetimes is that at least some of the parcels of air that are sampled have been through the deep branch of the BDC. By restricting comparisons to the polar stratosphere, it is primarily air from the deep branch of the BDC that is being investigated. Tracers, with a stratospheric sink, are most depleted from the deep branch because they have

had the most time for chemical loss to occur since they entered the stratosphere.

The comparison of the $N_2O$ and CFC-12 difference time series (Fig. 9) demonstrates that there is inter-annual variability that is consistent between the two gases in the Arctic. While the two species shown follow similar patterns over time, there appear to be larger differences in the CFC-12 comparison than in the $N_2O$ comparison. There are two possible (and related) reasons for this difference: the range in the concentrations of CFC-12 is much larger than that of $N_2O$, and there are differences

in their respective chemical losses. For example, the photolysis loss of CFC-12 is faster than that of $N_2O$ throughout much of the stratosphere, as evidenced by the differences in their lifetimes (102 years for CFC-12 and 123 years for $N_2O$). Generally, it appears that the model simulates higher concentrations of both species compared to ACE-FTS measurements through much of the stratosphere, with the largest differences occurring above 30 hPa. When the concentration of either tracer becomes very small (typically air that has descended from the upper stratosphere or mesosphere), the relative differences between ACE-FTS

and CMAM30HR can be enhanced. These differences are most clear in the $N_2O$ comparisons during the autumns of 2004 and 2009, and the springs of 2007 and 2010; while in the CFC-12 comparisons, the springs of 2005, 2006, and 2008 exhibit additional occurrences of this feature.

During each of these periods, ACE-FTS observed much lower concentrations compared to CMAM30HR. Both the speed and structure of the residual circulation within the CMAM30HR run can contribute to the observed differences. It is possible

that the BDC in the CMAM30HR simulation is drawing air through the deep branch of the BDC too rapidly. The vertical structure of the differences observed in Fig. 9, particularly between 70 hPa and 30 hPa, may be caused by ACE-FTS measuring a descent in the air mass that CMAM30HR does not simulate. It is more likely that the model circulation is not moving enough air through the loss region of these tracers and through to the polar vortex. For photolytic tracers, the structure of the circulation is more important than the speed of the residual circulation because photolysis rates are so fast. Above a certain level in the

stratosphere, the tracer is completely destroyed when air passes through the region. The distribution of photolytic species is a mixture of air that was passed through the region of rapid loss and the air that by-passed the loss processes. This result is consistent with the work of Pendlebury et al. (2015), where they found large differences in temperature and ozone between satellite observations and CMAM30.

There is less interannual variability in the Southern Hemisphere comparisons (Fig. 10) than that in the Northern Hemisphere

(Fig. 9). This is expected since the variability of the southern polar stratospheric dynamics is much less than that in the northern polar stratosphere. However, the magnitude of the differences between the measurements and simulations is larger in the Antarctic stratosphere than in the Arctic stratosphere. Moreover, while the patterns of the differences in $N_2O$ and CFC-12 are quite similar in Fig. 10a and 10b, the magnitude is more pronounced in the CFC-12 comparisons. The largest differences occur above 30 hPa where the concentrations of CFC-12 are extremely low. The peak in the magnitude of the CFC-12 differences

appears to increase in vertical extent through the Austral springtime. The largest differences tend to occur during summer





(December) at around 40 hPa for both tracers. The differences in CFC-12 are established at the top of the vortex in July and propagate down until vortex break up in December. However, this propagation doesn't occur to the same extent in the N$_2$O comparisons, which may be a reflection of the differences in the chemistry of the two tracers. For example, a source of N$_2$O in the lower thermosphere has recently been identified in ACE-FTS measurements by Sheese et al. (2016). The N$_2$O source descends into the mesosphere and stratosphere, thereby influencing air that is circulated in the BDC (Sheese et al., 2016). The transport of enhanced N$_2$O downwards from the upper atmosphere has also been detected by Funke et al. (2008a, b). The CMAM30HR does not include this source of N$_2$O. The results presented here are consistent with the methane comparisons discussed in Pendlebury et al. (2015). Generally, the descent of the model's high bias is observed in both hemispheres for all three trace gas species. The results of Pendlebury et al. (2015) indicate that the high bias is consistent with a fast BDC and that the downward propagation of the bias is a problem with the parameterizations in the model above 10 hPa.

## 5.2  The extratropical tropopause

The transport barrier at the extratropical tropopause can be permeable, allowing the exchange of air between the troposphere and the stratosphere. Understanding how CMAM30HR simulates this exchange assists in the interpretation of mixing effectiveness in the model and the impact of its vertical resolution on the comparisons. Since ACE-FTS predominantly samples the polar regions and has fewer samples of extratropical latitudes, interpreting latitudinal or seasonal dependence of the exchange of air across the tropopause in the full atmosphere using these measurements must be considered from a tropopause coordinate perspective (Hegglin et al., 2009). In this study, a diagnostic of the tropopause barrier has been developed for comparison between the simulations and the measurements. The tropopause height, used in this analysis to define tropospheric air and stratospheric air, is the thermally defined tropopause based on the derived meteorological products for ACE-FTS (Manney et al., 2007) and based on sampled temperature profiles from CMAM30 output for the CMAM30HR simulations.

Since CFC-11 has a strong vertical gradient in concentration in the stratosphere, it can be used as a proxy for determining the exchange of air across the tropopause. A data point is defined as an intrusion based on two criteria: the physical location of data point and the concentration relative to a tropospheric concentration threshold and only data below the 420 K potential temperature layer of the atmosphere have been considered. The tropospheric threshold was defined separately for each hemisphere as the tropospheric mean minus 1.5 times the tropospheric standard deviation. An intrusion frequency metric has been developed for comparison between the simulated and observed concentrations. A tropospheric intrusion is identified when a measurement is physically located in the stratosphere but its concentration is larger than the tropospheric threshold. A stratospheric intrusion is identified when a measurement is physically located in the troposphere but its concentration is smaller than the tropospheric threshold. For each five degree latitude bin between 20° and 60° latitude (N and S), the frequency of tropospheric and stratospheric intrusions have been determined. This technique was used to calculate frequencies for both the ACE-FTS measurements and CMAM30HR profiles.

The comparison of ACE-FTS and CMAM30HR tropospheric and stratospheric intrusions within the southern and northern extratropical latitudes is shown in Fig. 11a and 11b, respectively. There appears to be better agreement for the stratospheric intrusion comparisons and at some latitudes there is very good agreement between ACE-FTS (red) and CMAM30HR (black).





However, there are some latitudes, such as 30-40° S and 20-25° N, where the stratospheric intrusions identified in the simulation are a factor of two fewer than those from the ACE-FTS measurements. In general, isentropes that are in the extratropical lowermost stratosphere are in the troposphere in the tropics (e.g., Holton et al., 1995). Fast isentropic transport occurs because wave motions cause air to rapidly change latitude, leading to the transport of stratospheric air into the troposphere. Since the disagreement in measured stratospheric intrusions is largest at latitudes where this isentropic transport tends to occur implies that CMAM30HR is not capturing this mechanism well or simply lacks the resolution to fully resolve the stratospheric intrusions features.

Based on Fig. 11, tropospheric intrusions occur more frequently in the ACE-FTS data than stratospheric intrusions and vary more significantly in number across the latitudes in both hemispheres. Additionally, the differences between the measurements and simulations are larger than for the stratospheric intrusions. It is possible that there is more tropospheric air found in the stratosphere with this method than stratospheric air found in the troposphere because tropospheric CFC-11 is more easily distinguishable. The manner in which the intrusions have been defined here does not rule out stratospheric air being identified as tropospheric if there is rapid, poleward transport out of the lower tropical stratosphere. Schwartz et al. (2015) suggest that it is easier to distinguish tropospheric intrusions into the stratosphere using tropospheric tracers, and stratospheric intrusions into the troposphere using stratospheric tracers. Tropospheric CFC-11 concentrations found in the stratosphere at extratropical latitudes indicate air that likely has not cycled through the BDC because the air parcels have not experienced any loss processes that can only happen in the stratosphere; while stratospheric air in the troposphere could be representative of a range of aged air. For example, this air could have cycled through the BDC very quickly by re-entering the troposphere in the subtropics or it could have gone through the deep branch, allowing it to be more depleted and therefore more identifiable as stratospheric-like air.

A mechanism for extratropical tropospheric air to be uplifted into the stratosphere has been identified recently. Building on the work of Pan et al. (2009), Peevey et al. (2014) showed that the occurrence of double tropopauses is associated with the strength of the tropopause inversion layer (TIL), as well as Rossby wave breaking. They also showed that as the strength of the TIL increases, cyclonic circulation in the upper troposphere switches to anticyclonic circulation, thereby driving an increase in the upward motion. Based on the double tropopause calculations of Schwartz et al. (2015), it is likely that ACE-FTS observes this phenomenon of upward motion, leading to a higher frequency of tropospheric intrusions across the extratropical regions. CMAM30HR appears to inadequately simulate this mechanism since it doesn't have a sharply defined tropopause compared to reality (Birner and Bönisch, 2011). It is possible that both the spatial and vertical resolutions of the simulations performed limit the model's ability to capture this synoptic-scale activity.

Seasonal averages of the tropospheric and stratospheric intrusions are shown in Fig. 12, where Fig. 12a and Fig. 12b are the tropospheric intrusion frequencies for the southern and northern extratropics, respectively and Fig. 12c and Fig. 12d are the stratospheric intrusion frequencies for each hemisphere. For this case, the data shown in Fig. 11 were averaged seasonally in each hemisphere and the error bars represent the latitudinal variability defined as one standard deviation of the mean. Recall that since the intrusion frequencies in Fig. 12 are affected by the sampling pattern of ACE-FTS, the seasonality shown may not be representative of the actual seasonality of the atmosphere. The stratospheric intrusion comparisons are remarkably good;





there does not appear to be a significant difference between the simulations and the measurements. Therefore, it is unlikely that there is a physical mechanism or deficiencies in the model leading to the differences observed in Fig. 11 and the differences are primarily due to the model resolution. The comparisons of tropospheric intrusions exhibit similar behaviour between the two hemispheres. The measurement-model differences appear to be largest during SON and smallest during MAM. The consistency

of the increase during SON in tropospheric intrusion frequency between the two hemispheres may be the result of convective over-spill into the stratosphere across the subtropical tropopause barrier if the majority of the tropospheric intrusions are driven by convection in the tropics. This convective influence could extend across the extratropical latitudes via isentropic transport from the tropics. The strength of the convection during JJA and SON and its influence on mixing across the tropopause barrier is observed by the largest variability in the stratospheric intrusions. For example, if the model is does not have a strong enough

Asian monsoon circulation, then there would be fewer tropospheric intrusions simulated, particularly in JJA.

### 5.3   The tropical pipe

In this section, JPDFs are used to investigate the tropical pipe barrier. Figure 13 illustrates the $N_2O$/CFC JPDFs of the entire Northern Hemisphere stratosphere, beginning at 2 km above the tropopause to 30 km (up to 10 hPa), for ACE-FTS (Fig. 13a,c) and CMAM30HR (Fig. 13b,d) for CFC-12 (Fig. 13a,b) and CFC-11 (Fig. 13c,d). Hegglin and Shepherd (2007) have

highlighted the use of this technique for comparing CMAM and ACE-FTS measurements. The $N_2O$/CFC-12 JPDFs (Fig. 13a and 13b) exhibit a quasi-linear relationship in both the measurements and the simulations. The loss rates of CFC-12 and $N_2O$ are very similar in the upper troposphere and lower stratosphere, leading to a linear relationship in the JPDFs. The model JPDFs tend to peak at the higher concentrations of $N_2O$ and CFC-12 but are more evenly distributed throughout the range of concentrations. This difference between the measurements and simulations is likely due to the overly rapid BDC in the

model simulation, as previously discussed. The ACE-FTS data in Fig. 13a are much more scattered than the CMAM30HR in Fig. 13b, where the simulated tracers are highly correlated and the JPDF is highly compact. The differences in the spread of the correlations are related to differences in mixing and chemistry, but are also influenced by the precision of the ACE-FTS measurements and the constraint of the boundary conditions in the simulations. The CMAM30HR JPDF is very compact because of the similarity of the chemical losses of the two tracers in the model. Additionally, the surface boundary conditions

applied do not represent the variability observed in the atmosphere. It is the atmospheric variability that contributes to the variability observed in the ACE-FTS JPDF around 150-200 ppbv of $N_2O$.

    The $N_2O$/CFC-11 JPDFs show two segments of linear correlations in both the measurements and model results (shown in Fig. 13c and 13d, respectively) that would have otherwise been overlooked in data dense tracer-tracer correlation plots (also see Hegglin and Shepherd (2007)). The presence of a bimodal correlation between $N_2O$ and CFC-11 has been previously

observed by Plumb (1996). The separation is caused by differences in local chemistry in the tropical pipe region. The tropics are somewhat isolated from the midlatitudes so that the steeper slope is a signature of the local chemistry, or the relative loss rates of CFC-11 and $N_2O$ in the tropical lower stratosphere. This relationship is observed because the photochemical lifetime of CFC-11 is shorter than the time scale for mixing by horizontal eddy transport (Plumb, 2002). According to Plumb and Ko (1992), slope equilibrium conditions that define the linear relationship seen in the $N_2O$/CFC-12 JPDF are only satisfied if the



photochemical lifetime of a species is much greater than the horizontal eddy transport. When this condition is satisfied, the slopes of the isopleths are only a function of atmospheric circulation.

Isolating the measurements and simulations in the tropical region allows for the characteristics of the tropical pipe to be investigated. A JPDF comparison is provided in Fig. 14 for both the ACE-FTS measurements and the CMAM30HR simulations

in the tropics during the DJF, MAM, JJA, and SON seasons. The data were selected from the tropical latitude region using estimates of the turn-around-latitude, the height-dependent latitude where the tropical upwelling is zero, determined from CMAM30HR vertical velocities. Bimodal behaviour is observed in each season and in both the measured and simulated JPDFs. In general, the maximum of the JPDF appears to be positioned towards higher concentrations in the simulation compared to the measurements, where the maximum in the probability tends to extend throughout the shallower segment. As was observed

in the $N_2O$/CFC-12 JPDFs in Fig. 13, the probabilities tend to be weighted towards the higher concentrations implying that there is younger air (of tropospheric origin) in the simulated stratosphere than the measured stratosphere. Of note is that the length and the width of the shallower segment are consistently larger in the measurements. The longer length – extending to low concentrations – indicates the presence of older air not found in the model simulations. The larger width also coincides with the degree of separation between the primary and secondary segment in each season, where the simulated JPDFs appear to

have a much greater separation than the measurements. These features are likely dependent on the amount of quasi-horizontal mixing influencing the JPDFs, implying that there is not enough quasi-horizontal mixing occurring in the simulation. However, the differences between the measurements and simulations are primarily in the steepness of this segment of the JPDF, which is a sign of having not enough mixing into the tropics, rather than a product of a too rapid tropical ascent. If it was just too fast ascent, both tracers would be affected in the same way. But because the mid-latitudes $N_2O$/CFC-11 relationship is less

steep, it is the mixing in of this air that makes the tropics have shallower slope. Then it becomes a question of whether mixing is underestimated because the ascent is too fast or if the model doesn't simulate the structure of the pathways of the BDC correctly.

Figure 14 isolates the $N_2O$/CFC-11 JPDFs to the tropical region only, as defined by the turn-around-latitudes, for both the ACE-FTS measurements (left column) and the CMAM30HR simulations (right column) for each season. There is an evolution

of the characteristics of the JPDFs shown in this figure. During DJF, there is minimal separation between the two segments. The position of the maximum in the JPDF (the red region) in Fig. 14a does not extend below 200 ppbv of $N_2O$, while the location of the maximum of the simulated JPDF in Fig. 14b is limited to above 250 ppbv of $N_2O$ with the maximum primarily located above the separation of the two segments. During MAM, the maximum probability of the ACE-FTS JPDF in Fig. 14c extends throughout the shallow and steep segments. This implies that there is significant mixing occurring during this time

period. In the simulated MAM, Fig. 14d, the maximum of the probability is restricted to the higher concentrations of $N_2O$, prior to the separation of the two segments. This indicates that while mixing is occurring, it does not occur frequently enough to simulate the atmosphere well.

During JJA, the two segments begin to separate in both the measured and simulated JPDFs. The shallower segment in the simulated JPDF is the shortest during JJA of all the seasons with the maximum of the probability residing at the highest con-

centrations. This indicates that the upwelling during this season is too strong and it may also be too isolated since there appears



to be more mixing in the measured JPDF. During SON, the separation between the segments is most prominent, indicating that this time of year exhibits the least mixing. Based on these comparisons, it is still difficult to discern the relative contributions of the tropical upwelling and quasi-horizontal mixing to the differences in the JPDFs of the model and measurements. To interpret these differences, the results of the TLP model simulations described previously are used. These simulations provide a basis

for determining how much the residual circulation needs to slow down within CMAM30HR and what impact that may have on mixing of air between the tropical and extratropical regions.

## 6    Using a tropical leaky pipe model to interpret CMAM30HR

The TLP model is used here to identify the changes to the CMAM30HR tropical upwelling and effective mixing that may improve the simulations. It is difficult to adjust model parametrizations to modify wave breaking because many aspects of

the model would be impacted with no way of separating the effects (Ray et al., 2016). This includes the complexity of wave activity that contributes to stratospheric mean circulation and mixing. Therefore, without a simplified model, diagnosing the causes of discrepancies between measurements and model output solely from the global model is problematic. Prior to running the experiments used here, the TLP model was tuned to be representative of CMAM30HR by fitting estimates of the mean tropical upwelling ($w*$) and mean mixing levels ($\epsilon$). The tuning procedure is described in Ray et al. (2016). The same set of

TLP model experiments Ray et al. (2016) describe are also used here but the analysis has been modified to investigate the behaviour of the tropical pipe in CMAM30HR in further detail.

     A reduced mean circulation would likely correspond to less mixing and longer mixing times. Mixing levels are defined in the TLP by the ratio of horizontal mixing mass flux to horizontal mean mass flux scaled by the width of the tropical pipe region (Garny et al., 2014). Ray et al. (2016) found that the CMAM30HR simulations best match the ACE-FTS measurements when

the $w*$ is between 0.27 mm/s and 0.32 mm/s (a reduction from the fitting estimate of 0.4 mm/s) and $\epsilon$ ranges from 0.7 to 1.2 (an increase from the fitting estimate of 0.55). Based on the in-mixing time profiles shown by Ray et al. (2016), it is apparent that the CMAM30HR mixing times need to increase to slow down the mixing at all levels, although the differences are only significant in the lower part of the stratosphere, below 20 km, and above 24 km. These are physically consistent changes since the mean circulation is driven by wave breaking, which also causes mixing between the tropics and extratropics.

In this section, the TLP simulations are used in a spatial and seasonal context to determine the changes required in the CMAM30HR simulations to match the ACE-FTS observations. The three regions investigated (the tropics and the northern and southern extratropics) were defined by the turn-around latitude of the tropical upwelling and by exclusion of the polar vortex in each hemisphere using a 1.2 x $10^{-4}$ s$^{-1}$ scaled potential vorticity threshold (e.g., Manney et al., 2007). The vortex is excluded in these comparisons because the TLP model does not simulate its complexity. For each of the 480 simulations

of the TLP model and the ACE-FTS measurements, profiles for CFC-11 and CFC-12 were averaged over 16-27 km in the extratropics and 16-29 km in tropics. To capture the monthly coverage of the ACE-FTS measurements, a weighting function for each region and season was applied based on the relative contribution of the occultations observed in each month of the particular season and region (the tropics and northern/southern extratropics). The vertically averaged and monthly weighted




measurements were compared to the individual simulations of the TLP model that represent varying levels of tropical upwelling and mixing efficiencies. For each of the comparisons, the absolute value of the differences is used and scaled to the maximum of the range of all differences so that each of the scales range from zero to one.

Figure 15 shows four examples of these comparisons plotted by the seasonally averaged, regionally specific modelled values of mixing, $\epsilon$, and upwelling, $w*$, for each simulation. The x-axis and y-axis values do not represent the setting used to run the simulation - they are the values simulated by the TLP model and lead to the curvature seen in the plots. The shading of each plot indicates the level of agreement, where the darker regions indicate the minimum differences between ACE-FTS and the TLP simulations. The white contours illustrate the agreement isolines. The relative CMAM30HR position is shown by the red marker and the error bars represent the estimated range of uncertainty based on the optimization exercise with the TLP model. These comparisons all indicate that the CMAM30HR values of $\epsilon$ and $w*$ would require some change to bring the simulations closer to agreement with the ACE-FTS measurements (i.e. bringing the red marker towards the dark shaded region). The dependence of $w*$ on $\epsilon$ and vice versa is not the same across regions and seasons for either CFC-11 or CFC-12. There are often limited ranges in which agreement between the TLP simulations and ACE-FTS can be assessed but it is clear that changes could lead to improvements in the agreement (such as in Fig. 15b). There are some scenarios where the mixing is required to change much more than the tropical upwelling (see Fig. 15c for an example) or where there can be a range of mixing levels and upwelling values that lead to better agreement (as in Fig. 15d).

To quantify the changes that would bring CMAM30HR into agreement with ACE-FTS based on the TLP simulation comparisons, an agreement matrix has been calculated. This was done by finding where the differences between ACE-FTS and the TLP simulation are below a certain threshold (0.65 for tropics and 0.2 for extratropics) and determining the ranges of $w*$ and $\epsilon$ over which this occurs. The average values (square markers) and ranges (error bars) of the changes to $w*$ and $\epsilon$ calculated are shown in Fig. 16. The changes in $w*$ were calculated based on the absolute value of $w*$ so that the interpretation of the differences calculated was not dependent on the sign of $w*$. This means that a positive change is always an acceleration of the BDC, even when $w*$ is typically negative, such as in the extratropics. Figure 16a shows these changes for each region and across the four seasons. The sign of $w*$ values is indicated above Fig. 16a. The tropical region changes required are typically centered around zero for all seasons and have the largest range of changes of any region in any season, indicating that there is no argument for changes to the tropical upwelling in CMAM30HR. However, there are significant changes identified in three of the four seasons in the extratropical region of the Northern Hemisphere, where the range in the change does not include zero. In DJF, an increase in $w*$ is required in the Northern Hemisphere, implying that strength of the downwelling during this season should increase by up to 0.08 mm/s. There is a similar requirement during MAM in the Northern Hemisphere with a minimal range, requiring an increase in the downwelling by approximately 0.05 mm/s. By summer in the Northern Hemisphere (JJA), the tropical pipe has shifted poleward such that the Northern Hemisphere extratropical $w*$ values are positive and require a decrease in value as compared to the measurements. This season tends to have the most active tropical upwelling but these results suggest that the upwelling in CMAM30HR maybe too rapid or, since the southern extratropics appear to have increased downwelling during JJA, may be displaced in latitude.



The calculations of changes to the mixing parameter, $\epsilon$, provide more clarification, particularly for the tropical region. Figure 16b shows that tropical mixing needs to increase in every season. There appears to be a seasonal cycle in this result, where the maximum occurs during JJA. The preceding season remains at approximately the same average change required as DJF but the range is much more narrow leading up to the large increase in JJA and subsequent drop off in SON. This result

quantifies and supports the idea that the difference in the separation of the JPDF segments between CMAM30HR and ACE-FTS is due to insufficient quasi-horizontal mixing between the tropics and extratropics. In the Northern Hemisphere, there are significant changes proposed in both MAM and JJA. In particular, JJA appears to require increased mixing in all regions studied, implying that there is a substantial deficiency in the CMAM30HR simulation during this season. The most significant physical mechanism for mixing during this season is the Asian monsoon. The quality of this transport mechanism has not been

directly assessed in CMAM30. It is unclear as to whether the mechanism has a direct or indirect effect but the Asian monsoon is a prominent climatological feature of the upper troposphere and lower stratosphere at this time of the year and it can be speculated that the required additional mixing may be related to the strength or extent of the simulated monsoon.

## 7 Conclusions

In this work, ACE-FTS measurements of CFC-11, CFC-12, and $N_2O$ have been used to assess the CMAM30HR simulations of

these tracers and, thereby, indirectly the transport processes in the lower stratosphere in the model. By treating each tracer in the specified dynamics simulation explicitly, the CMAM30HR run allows for the direct comparison of the measurements to model output. The advanced sampling technique employed here allows for detailed interpretation of the comparisons. Of the species investigated, it was found that CMAM30HR consistently overpredicts tracer concentrations in the lower to mid-stratosphere. The largest and most widespread overpredictions occur in the Northern Hemisphere winter and spring, when the BDC is most

active in that hemisphere.

The investigation of simulated mixing barriers identified a number of issues in the CMAM30HR simulations. The polar vortex comparisons reveal issues in both the timing and strength of the downwelling portion of the deep branch, which is likely directly related to the too-rapid overturning nature of CMAM's BDC. The extratropical tropopause barrier in the model appears to represent stratospheric intrusions well, as evidenced by Fig. 12c and 12d. However, tropospheric intrusions are

poorly simulated in most seasons (Fig. 12a and 12b), with the largest discrepancies occurring during JJA in both hemispheres. The tropical pipe mixing barrier analysis suggests that while the strength of the simulated BDC (i.e. upwelling in the tropics and downwelling in the extratropics) may partially explain the too young air found in the mid-stratosphere, mixing may play at least as prominent a role and seems to be underestimated particularly, in the JJA season in all regions.

Insufficient mixing during JJA may be related to the poorly simulated tropospheric intrusions during the same season and

this same issue may be directly related to the younger air in CMAM30HR. Garny et al. (2014) found that, in the subtropical lower stratosphere, younger air is the result of a combination of a speeding up of the overturning circulation and weaker mixing or recirculation of the stratospheric air between the tropics and midlatitudes. This may be evidence for insufficient mixing in the specified dynamics simulations being the cause of a too-rapid BDC. It is important to scrutinize the mixing levels in CCMs





and GCMs since it appears to be related to the mechanisms driving the projected trends in stratospheric circulation, thereby influencing the simulations of stratospheric ozone recovery and climate change. The techniques used in this work, including the advanced sampling and use of the tropical leaky pipe model, have proven illuminating. It is suggested that other CCMs and GCMs investigate the use of these techniques in future studies.

5  *Acknowledgements.* This project was funded by grants from the Canadian Space Agency (CSA) and the Natural Sciences and Engineering Research Council of Canada (NSERC). The Atmospheric Chemistry Experiment (ACE), also known as SCISAT, is a Canadian-led mission mainly supported by the CSA and the NSERC. The development of the CMAM30 data set was funded by the CSA. We thank Peter Bernath for his leadership of the ACE mission. We also thank Ted Shepherd, Dylan Jones, and John Scinocca for their leadership and support of the CMAM30 Project. The HATS measurements were funded in part by the Atmospheric Chemistry Project of the National Oceanographic and
10  Atmospheric Administration (NOAA) Climate and Global Change Program.





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

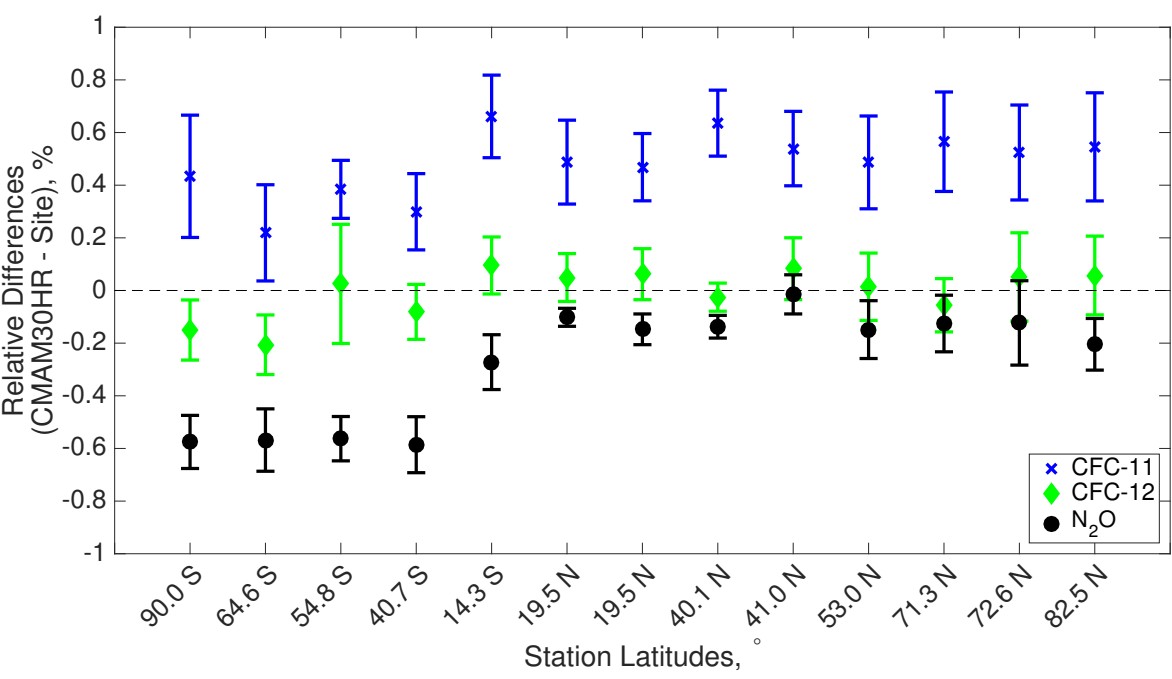

**Figure 1.** Comparison of CMAM30HR run to surface measurements, relative differences calculated as the site subtracted from the CMAM30HR simulation, divided by the average of the two, as described in the text. The differences and the uncertainties included are the mean and standard deviation of relative differences over the time series.



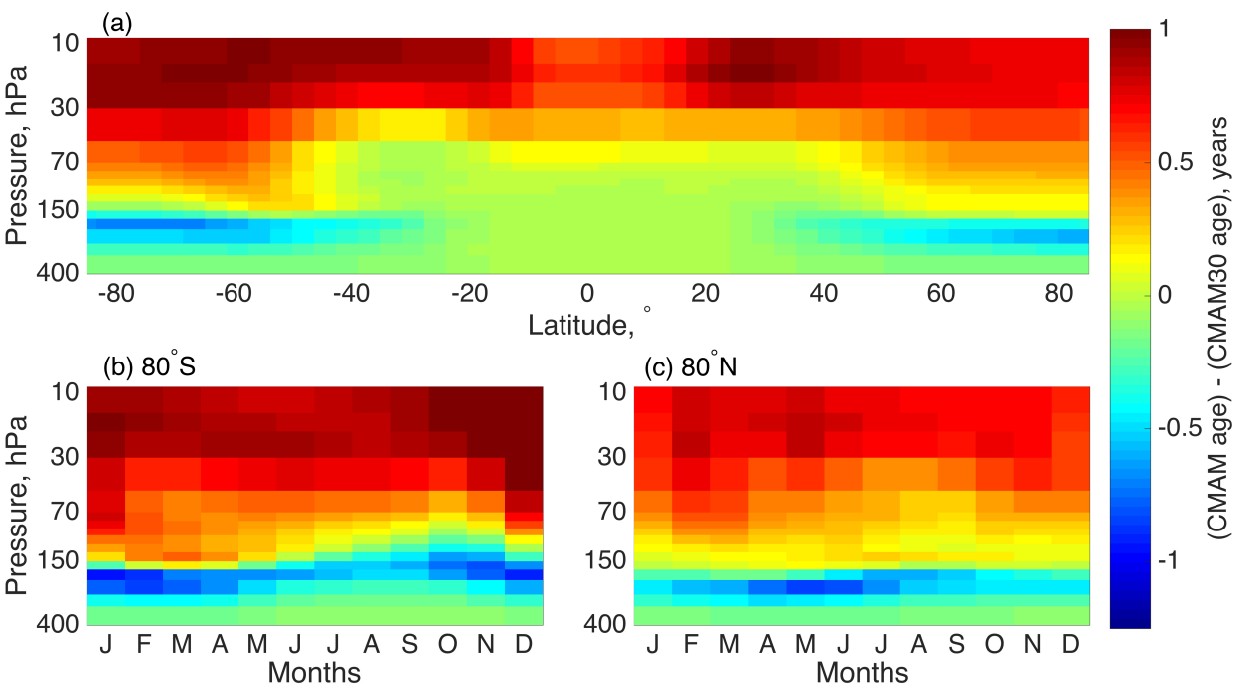

**Figure 2.** A comparison of the age of stratospheric air in the free-running CMAM and the nudged CMAM, CMAM30, averaged between 2004 and 2010. (a) The zonal mean difference of the CMAM30 mean age subtracted from the free running mean age (years), the monthly time series difference (b) at 80° S, and (c) at 80° N.





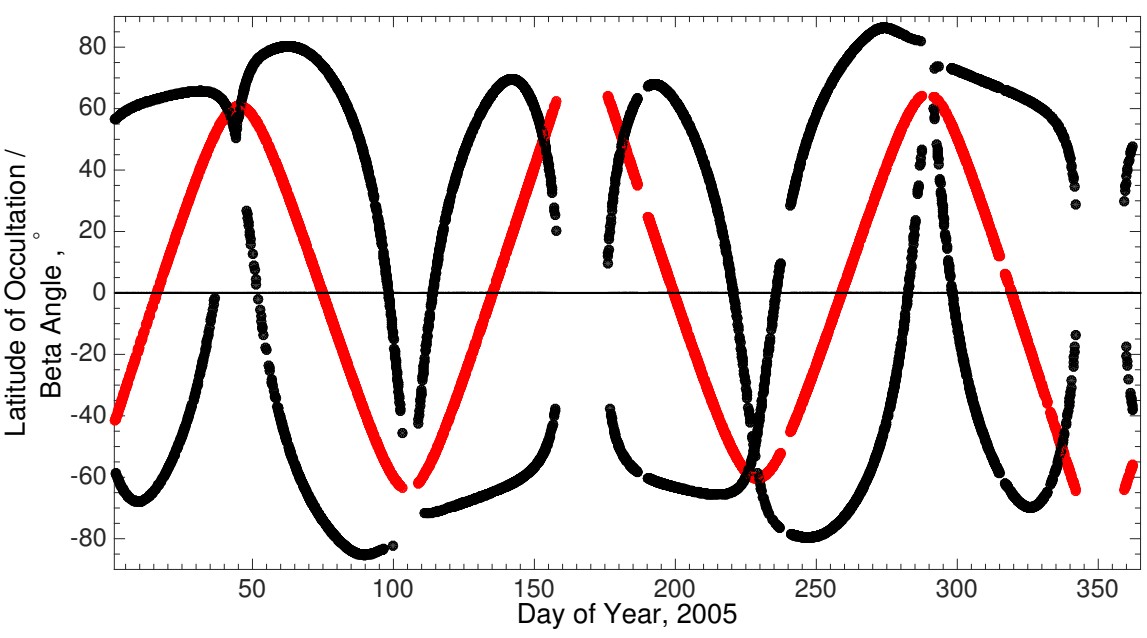

**Figure 3.** The ACE-FTS sampling pattern for the year 2005. Each black circle is the latitude of the 30 km tangent height of an occultation and each red circle is the corresponding beta angle of the occultation.

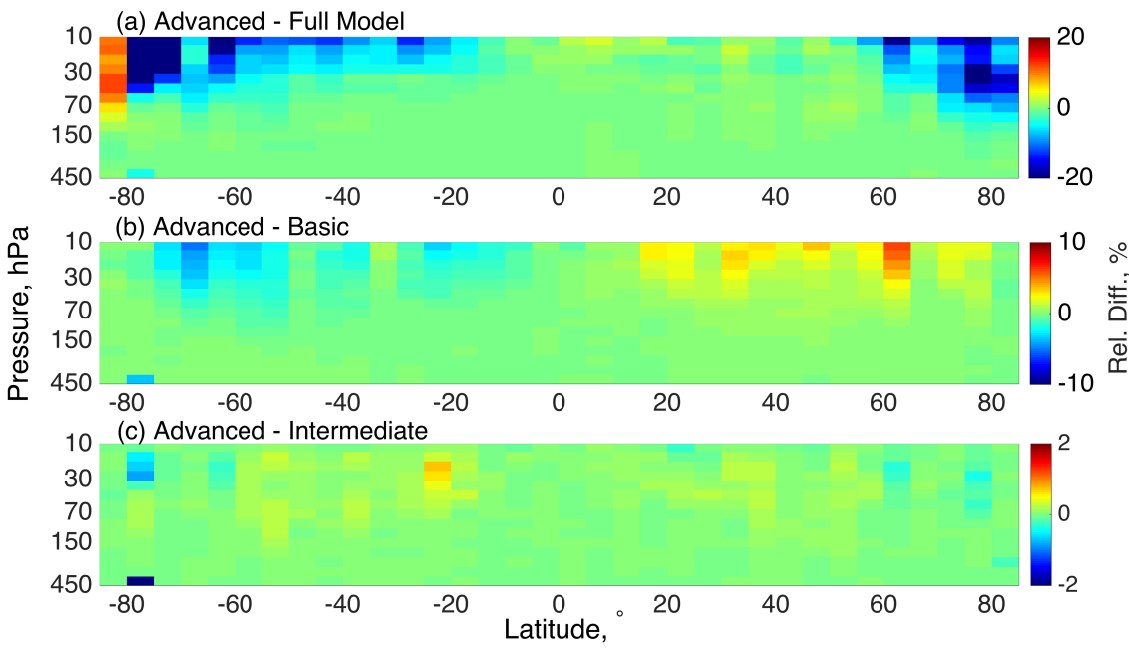

**Figure 4.** The comparisons of the three sampling methods described in the text using $N_2O$ simulations in CMAM30HR. The relative differences (%) defined as the difference between the advanced sampling and (a) the full model output, (b) basic sampling, (c) intermediate sampling. Note the different color scales in each panel.





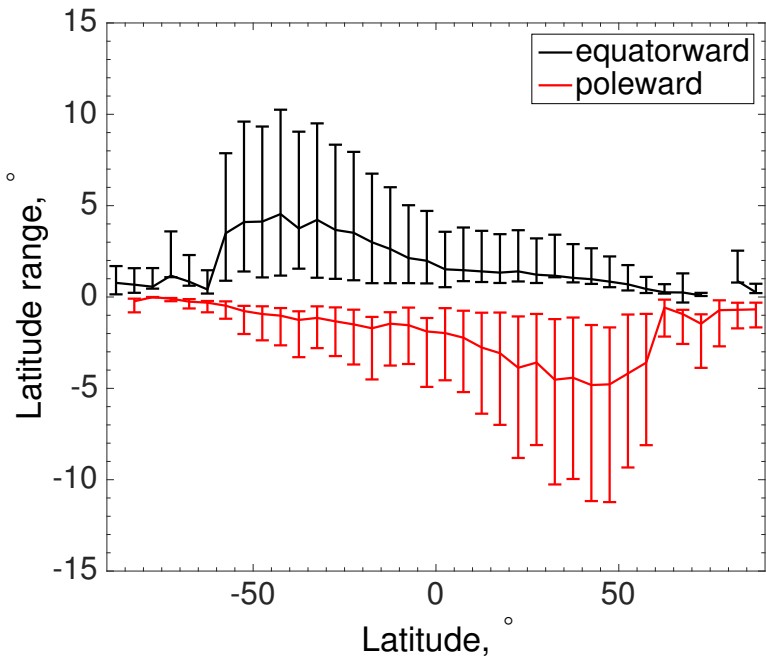

**Figure 5.** Average latitude ranges covered by ACE-FTS occultations in a given 5° latitude bin, separated by an equatorward bias (black) and a poleward bias (red) as defined in the text. The error bars indicate one standard deviation from the mean latitudinal extent for the given 5° latitude bin.



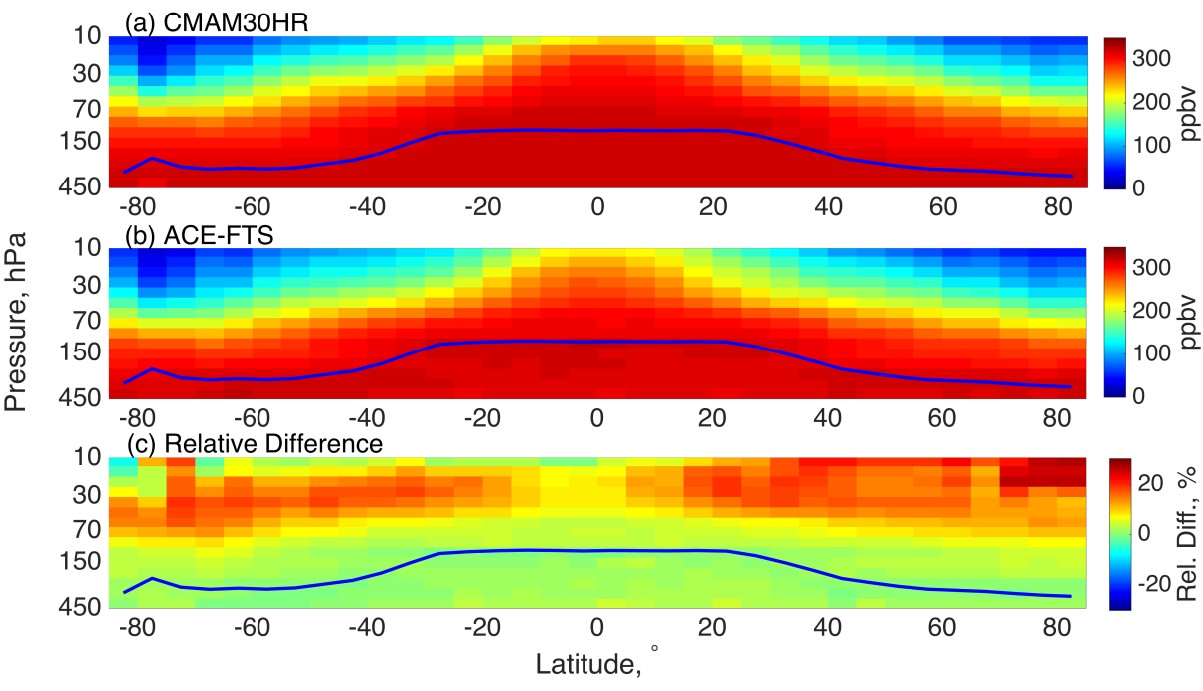

**Figure 6.** Zonally averaged latitude-altitude distributions of N$_2$O; (a) CMAM30HR (ppbv), (b) ACE-FTS (ppbv), (c) the mean relative difference (in %) between sampled model and ACE-FTS profiles, divided by the mean of their respective values (100*(CMAM30HR$-$ACE-FTS/((CMAM30HR$+$ACE-FTS)/2))). The blue line indicates the location of the thermally-defined tropopause.





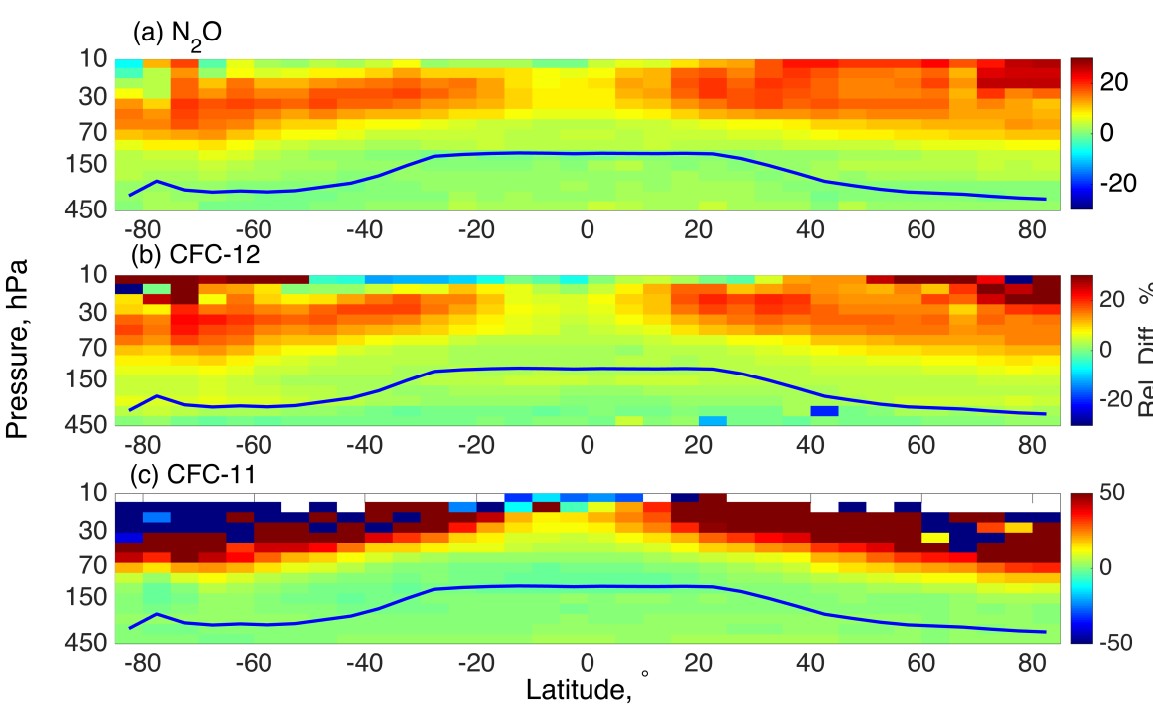

**Figure 7.** Same as Fig. 6c for (a) $N_2O$, (b) CFC-12, and (c) CFC-11. Note the different scale used on panel (c).





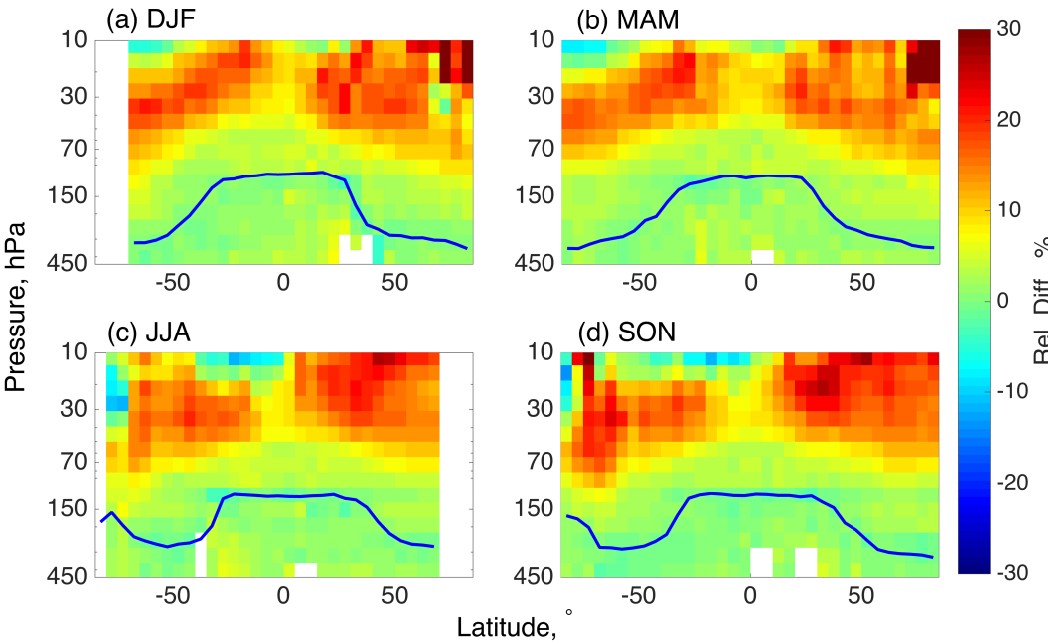

**Figure 8.** The relative mean of individual ACE-FTS profiles subtracted from CMAM30HR profiles of $N_2O$, divided by the mean of ACE-FTS and CMAM30HR for each season (a) December-January-February (DJF), (b) March-April-May (MAM), (c) June-July-August (JJA), and (d) September-October-November (SON).





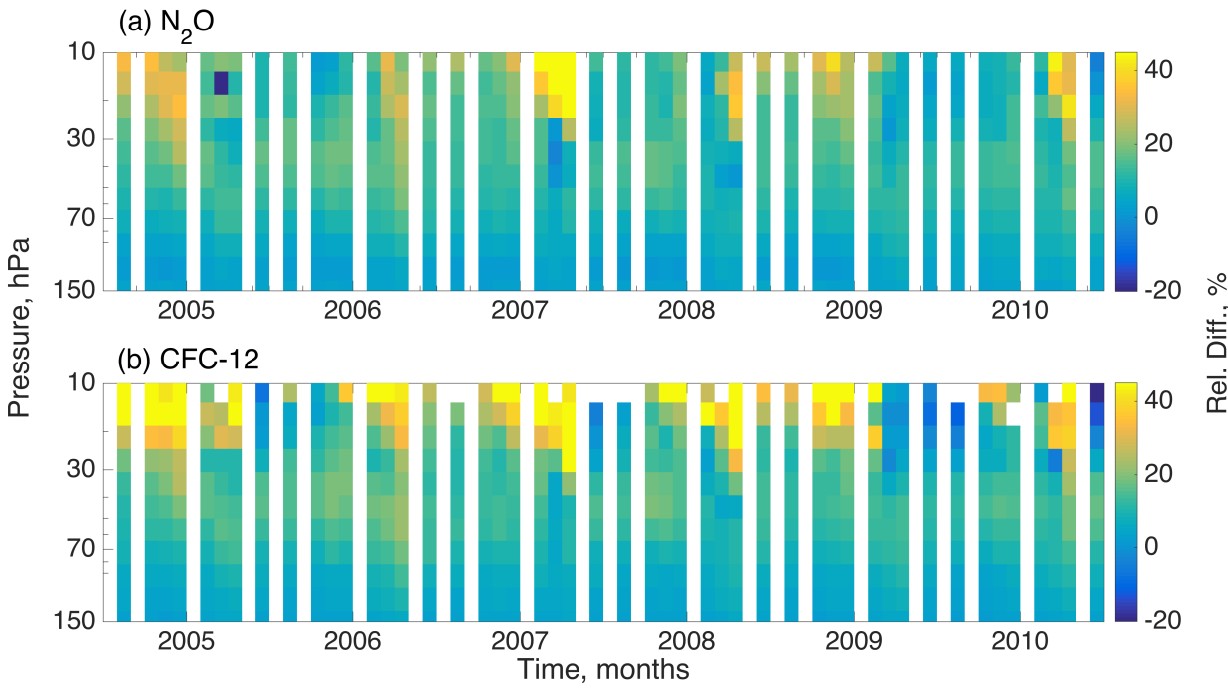

**Figure 9.** A monthly time series of the average relative differences between the model and the measurements (CMAM30HR minus ACE-FTS, divided by the mean), as in Fig. 6c for (a) $N_2O$ and (b) CFC-12 in the Northern Hemisphere (60° - 90° N) between June 2004 and May 2010.

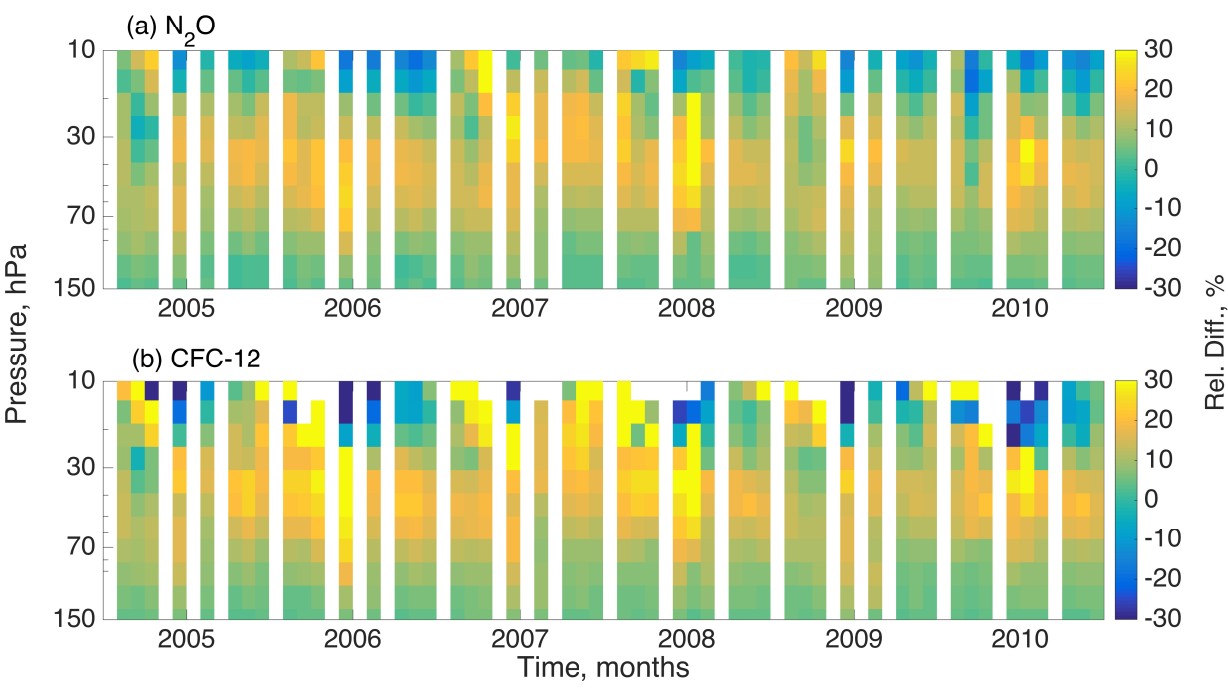

**Figure 10.** A monthly time series of the average relative differences between measurements and the model (CMAM30HR minus ACE-FTS, divided by the mean), as in Fig. 6c for (a) N$_2$O and (b) CFC-12 in the Southern Hemisphere (60° S - 90° S) between June 2004 and May 2010.





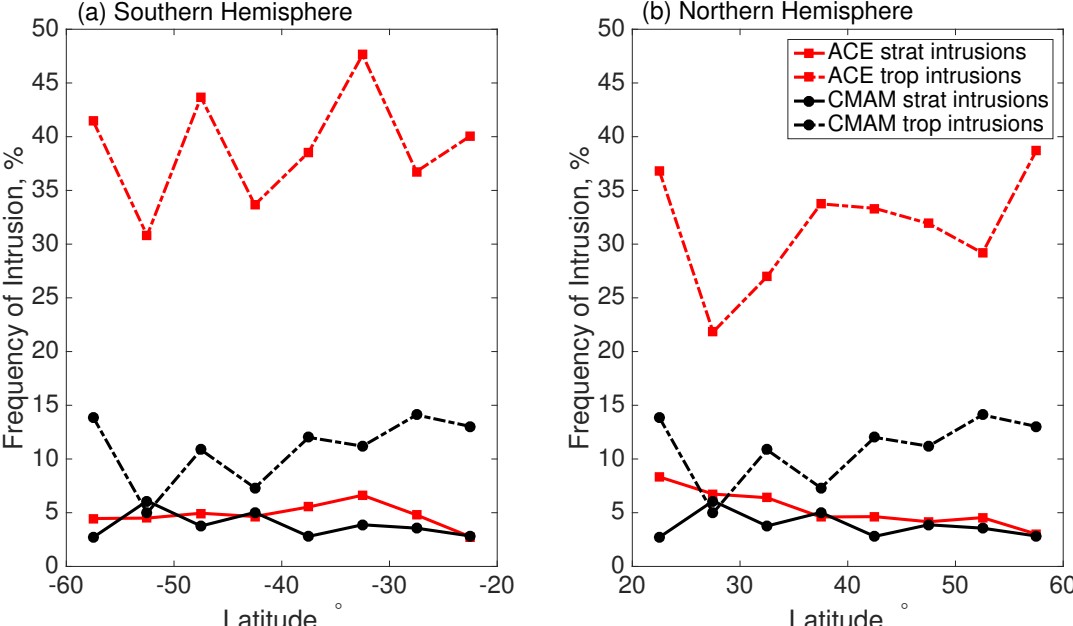

**Figure 11.** The frequency of intrusions across the tropopause for both CMAM30HR (black circles) and ACE-FTS (red squares) below 420 K in (a) the Southern Hemisphere and (b) the Northern Hemisphere extratropical region between 20° - 60° N/S, respectively. The intrusion frequencies are separated by stratospheric intrusions (solid lines) and tropospheric intrusions (dashed lines), as defined in the text, for each 5° latitude bin.




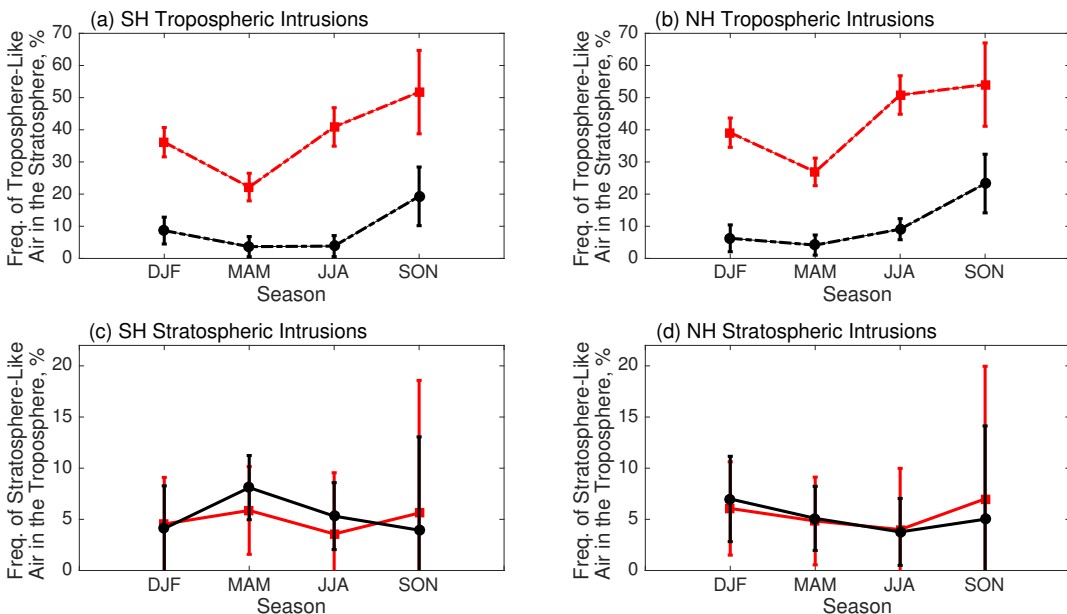

**Figure 12.** A seasonal representation of the intrusions depicted in Figure 11: (a) tropospheric intrusions in the Southern Hemisphere, (b) tropospheric intrusions in the Northern Hemisphere, (c) stratospheric intrusions in the Southern Hemisphere, and (d) stratospheric intrusions in the Northern Hemisphere. Each season is an average of the extratropical intrusion frequency with error bars indicating one standard deviation of the seasonal mean. ACE-FTS intrusions are in red squares and CMAM30HR intrusions are in black circles. Note the seasonality represented may be impacted by the sampling of ACE-FTS and may not representative for the full atmosphere





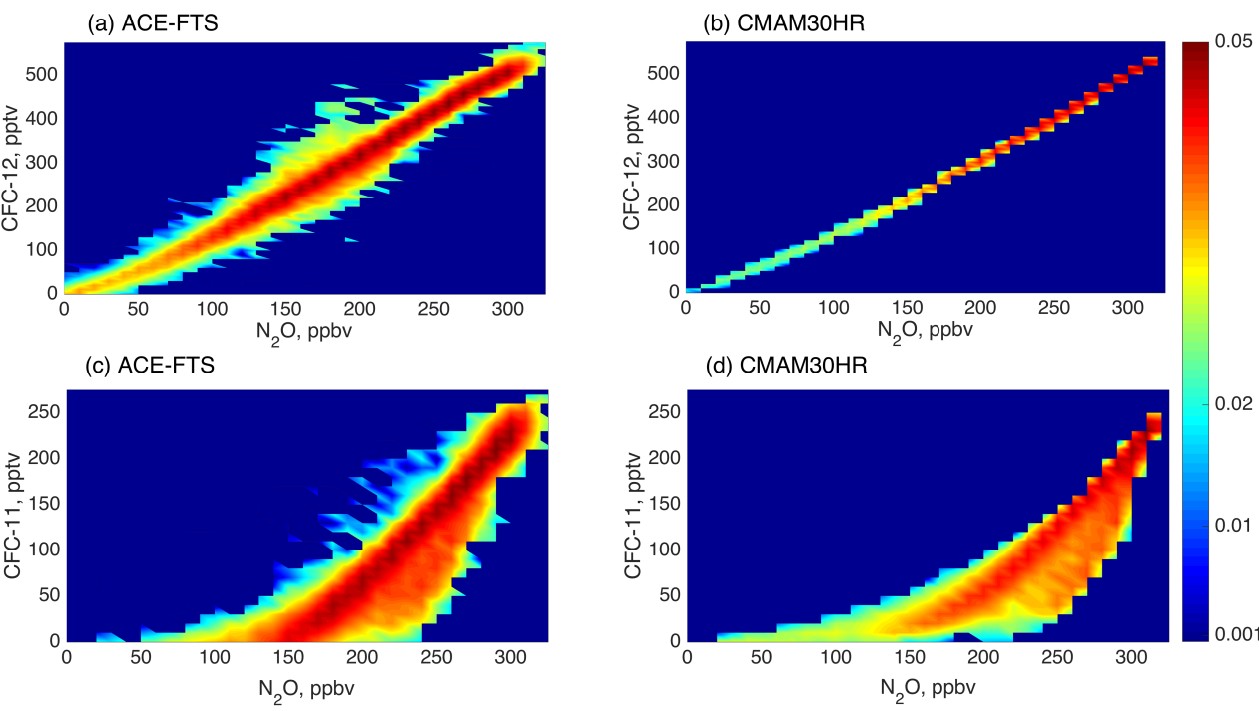

**Figure 13.** JPDFs, as described in the text, of $N_2O$/CFC-12 for (a) ACE-FTS and (b) CMAM30HR, and $N_2O$/CFC-11 for (c) ACE-FTS and (d) CMAM30HR. All stratospheric ACE-FTS observations and subsampled model output in the Northern Hemisphere are included.





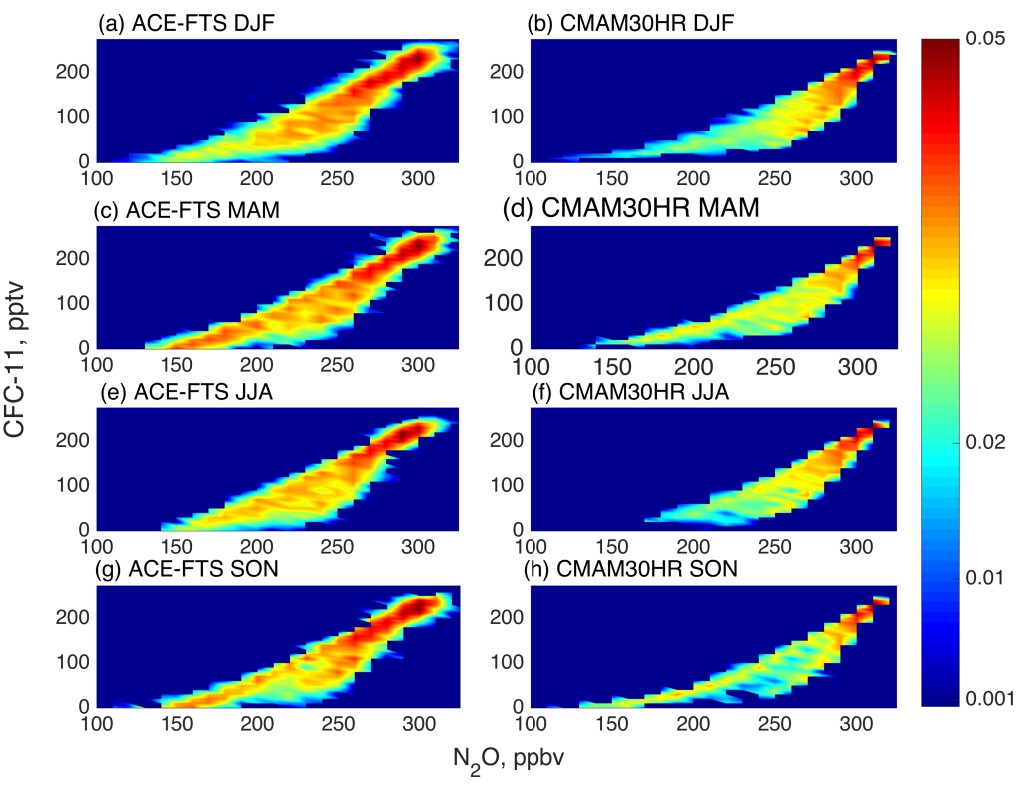

**Figure 14.** Tropical JPDFs of $N_2O$/CFC-11, as described in the text, separated by season. Only stratospheric ACE-FTS observations (left column) and CMAM30HR simulations (right column) within the height-dependent tropical turn-around-latitudes have been included.





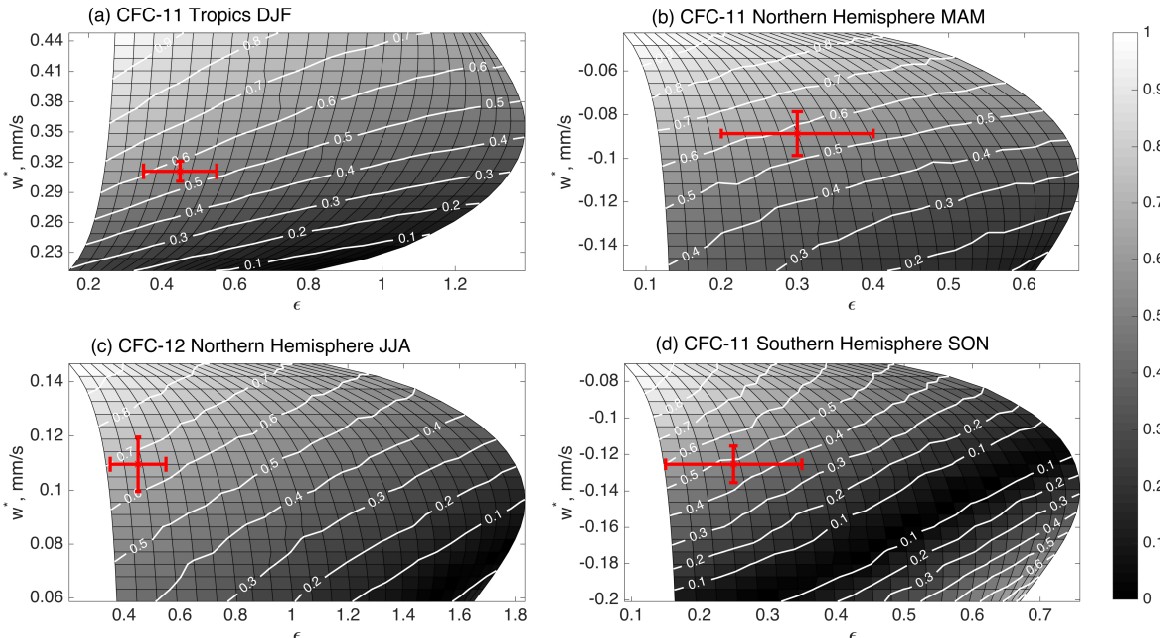

**Figure 15.** Examples of the TLP comparisons described in the text. The grey scale on each panel indicates the comparison between ACE-FTS and the TLP simulations over various $w^\star$ (y-axes) and $\epsilon$ cases (x-axes). The white lines identify the agreement with ACE-FTS (gray colour scale). The red marker indicates the CMAM30HR estimated values of $w^\star$ and $\epsilon$. The examples shown are (a) CFC-11 tropics during the DJF season, (b) CFC-11 Northern Hemisphere during the MAM season, (c) CFC-12 Northern Hemisphere during the JJA season, and (d) CFC-11 Southern Hemisphere during the SON season.



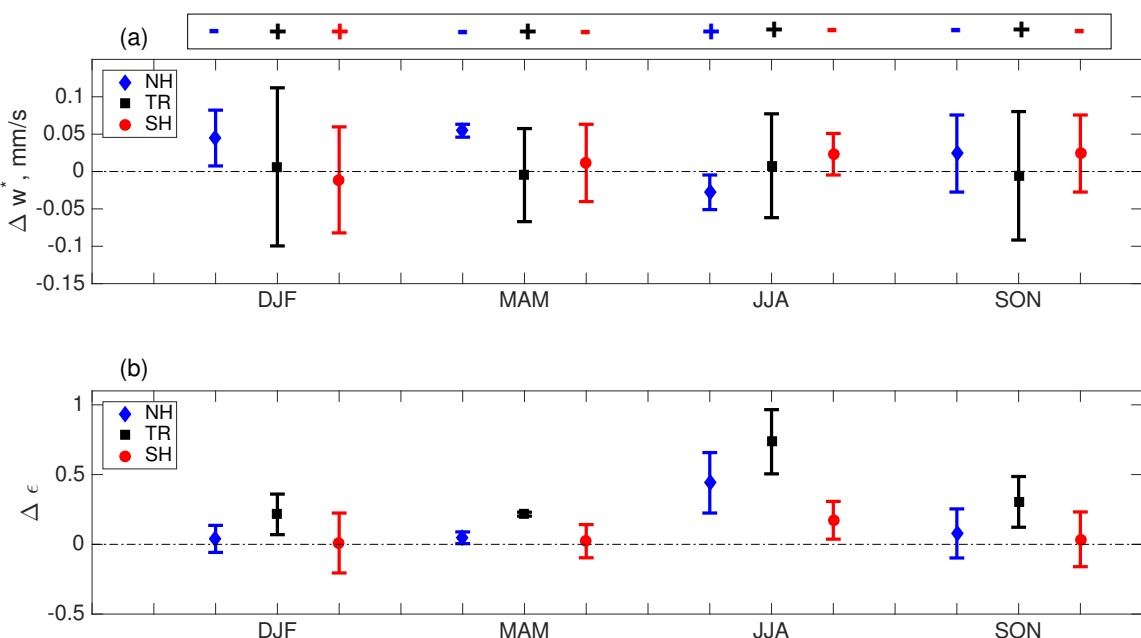

**Figure 16.** A summary of the TLP comparisons by season and region: (a) indicates the changes required in tropical upwelling, $w^\star$, while (b) indicates the changes in the mixing, $\epsilon$, required for the CMAM30HR simulation to agree with the ACE-FTS observations. The '-' and '+' symbols at the top of the figure indicate the direction of mass transport (downwelling or upwelling, respectively) for each region and season as indicated by color and location on the figure.