# Peer review of "Assessing stratospheric transport in the CMAM30 simulations using ACE-FTS measurements"

_Atmospheric Chemistry and Physics, 2017_

## Referee Comment (RC1) · Anonymous Referee #1 · 17 Jun 2017

The authors compare free-running and nudged simulations using the CMAM middle-atmosphere model to ACE-FTS measurements of long-lived tracers (

The analysis is generally well grounded and based on established analysis techniques, such as tracer-tracer correlation plots. Generally my impression is that the paper takes in a lot of information, making this a fairly dense read. For the future, I recommend to the lead author to break up such works into into smaller, separately publishable pieces. I don't think it would be adequate to recommend this course of action for the present paper as this is only a matter of presentation. The captions of some figures could be more detailed; for examples see below. In the comparison of models versus satellite measurements, a more thorough discussion of the effect of measurement uncertainty on tracer-tracer plots would be desirable. For example, the density plots of $N_2O$ versus

CFC-11 and CFC-12 (figures 13a and c), in the case of the satellite, are probably affected by measurement noise giving the JPDFs a fuzzy appearance. Such noise is absent in the model, making for a skewed comparison. Possibly averaging kernels of the ACE-FTS measurements could be used to define random noise to be added to the model data, making them more comparable to the measurements. If this is not practical, at least some text to this effect would be good to have.

Similarly, the discussion of differences in stratosphere-troposphere intrusions / extrusions mentions that resolution might factor into this comparison. At least for the detection of such structures in the data, this can be accounted for by removing small scales from the satellite data using a low-pass filter. Then the two datasets are nominally at the same resolution. This would however not address that the simulation of cut-off systems is fundamentally sensitive to numerical diffusivity in the model, causing reduced incidences of such systems.

Regarding the differences in age-of-air between the free-running and nudged version of the model, my impression is that this is partly caused by mass non-conservation in the nudging fields, whereby artificial divergence caused by relaxation towards reanalyses causes noise in the vertical motion fields. The effect of this might be increased numerical diffusion and a reduced age. Since CMAM is based in a spectral dynamical core, one could consider, in separate experiments, to only nudge divergence or only vorticity, to try to control this behaviour.

On the whole, the above amounts to a recommendation to publish after a minor revision. Minor comments:

P3L3: This sentence reads a little awkwardly – modelling and observations are independent activities. How about "The BDC is well characterized in models but remains poorly constraint in obs" or so?

P7L10: It's certainly possible to rescale the fields to construct approximations for the other tracers. But this requires further assumptions.

P7L20: Worth mentioning / discussing Meinshausen et al., Geosci. Model Dev., 2017 here. They have constructed boundary conditions for CMIP6 simulations that follow very similar ideas.

P8L6: It remains a little unclear to me how you can have systematic differences between the LBCs used to constrain the simulations and the long-term observations, when the obs were used to construct the LBCs.

P8L27ff: Perhaps not drag but noise in the vertical motion. The $\omega$ fields in the nudged and free-running model might show some differences.

P14L10: Perhaps insert "annual-mean" and some time information here (which period does the average represent?) Likewise in the caption, here and elsewhere.

P20L3: Here's where the above comment on model resolution applies. The key difference is not that the two fields are at different resolutions (that could be easily fixed) but that the finite resolution of the model leads to differences in the formation and lifetimes of the cut-off systems.

P34: More detail in the caption please. Which species, which network, which measurement principle, why are there these systematic differences when the measurements had been used in constructing the LBC for the model?

P46: Here's where I think measurement uncertainties make this a skewed comparison. The model output would ideally be folded with the averaging kernels and a-priori assumptions used in the retrievals of the ACE-FTS measurements before comparison with those measurements.

---

## Referee Comment (RC2) · Anonymous Referee #2 · 20 Jun 2017

This manuscript describes a detailed set of comparisons between specified dynamics simulations with the CMAM chemistry climate model and satellite-based observations of stratospheric long-lived tracers, with the goal of identifying discrepancies and therefore errors in the simulated stratospheric dynamics. The topic is suitable for ACP, and the conclusions reached by the authors are generally reasonably well supported by the analysis presented.

General comments:

1. I have no doubt that the "advanced" comparison technique–which samples the model data along the actual line of sight of the satellite instrument–is the best way of minimizing sampling errors in the CMAM vs. ACE-FTS comparison. But, the explanations given in Sec. 3.3 don't quite make sense to me. If the difference between the

advanced and intermediate methods is so small, as shown in Fig. 4c, this implies that the "line of sight" sampling is actually making very little difference to the sample means. The difference between advanced and basic (fig 4b) is much larger, which means that the most important source of sampling error has to do with a bias in the distribution of samples within each 5deg latitude bin (which is fixed by performing the 2D horizontal interpolation rather than using the closest neighbor gridpoint). It's possible these issues would be easier to sort out if Fig 4 showed differences between the 3 methods and the full model sampling, rather than differences between the advanced sampling and the other sampling methods. Or perhaps BASIC-FULL, INTERMEDIATE-BASIC, and ADVANCED-INTERMEDIATE. In any case, I believe the logic of the explanations here could be sharpened.

2. No doubt there is much more information given in the Ray et al. (2016) paper, but Sec. 6 requires a little more guidance on the set up of the TLP simulations. At some point, in passing, we learn that there were 480 TPL simulations, but it is not said how these simulations differ; presumably certain input assumptions are varied in the different simulations, but not, it seems, w* and epsilon directly. Also, the terms w* and epsilon should be better defined. At some point, w* is introduced as a mean tropical upwelling, but it is later used to quantify vertical motion in the extratropics. It's also not really clear if epsilon is a prognostic or diagnostic variable, and how it depends on height, latitude, time, etc.

3. I have trouble following the logic from Figure 15 to 16. Figure 15 seems to show that best agreement with the ACE-FTS measurements is achieved running the TLP model with parameter settings which produce the smallest w* and largest epsilon values (at least in Fig 15a,b,c). In fact, it seems that the range of TLP simulations is not large enough to find the actual best agreement with ACE-FTS–a point which could be discussed. But then, in Figure 16, it is implied that e.g., best agreement with ACE-FTS is produced with no significant change in w* values in the tropics. Something seems inconsistent here.

[Figure]

4. For many of the difference contour plots, it would be helpful if the colorbars were chosen such that positive differences could be more easily differentiated from negative differences. An example is Fig 6, where it is very difficult to know whether the CMAM-ACE differences in the UTLS are positive (like the middle stratosphere differences) or negative.

Specific comments

P1, l11: "The model consistently..." could be taken out of context–this conclusion is specific to the trace gases examined in this study (and probably wouldn't apply to ozone, for example).

P1, l14: the "too little isentropic mixing" should probably be connected if possible to a height or range of heights.

P2, l1: I'm not sure this is the only reason for the increase in interest in stratospheric transport–and it's a bit of a chicken and egg problem.

P2, l4: *Accurate* projections rely on good models.

P2, l5: Definitely the distribution of long-lived trace gases depends on the BDC... for short-lived species it may not have that much influence.

P3, l14: "has" or "have"? The word choice depends on whether the models project changes in the BDC (plural) or evidence of those changes (singular).

P4, l31: ACE-FTS measurements have high vertical resolution, not the instrument itself.

P8, l2: For comparisons of model to measurements, it's probably more intuitive to treat the measurements as the truth, and show relative differences of the model with respect to the measurements, rather than the mean of the measurements and model. It's of course not a big deal as long as it is clear how the calculation is being done, but I feel the simpler the calculation, the easier it is to interpret.

P8, l8: this statement of significance applies to the 1 sigma confidence level. For 2 sigma, I guess all differences would be not significantly different from zero.

P9, l24-26: Apparent contradiction between "increased isentropic mixing" and "slower shallow branch".

P12, l19: the end of this sentence could be misconstrued, as of course there has been vertical interpolation applied in the translation to the vertical levels of the ACE-FTS retrievals. I would suggest to remove this last part, and write "... location of the tangent points with altitude".

P13, l25: But the INTERMEDIATE sampling technique is also limited to the 30 km tangent altitude, and the differences to the advanced method are much smaller. Indeed, by construction, differences between a method using the variable tangent heights and one using only the 30 km tangent height should be zero at 30 km (which appears to be the case in the advanced-intermediate comparison). Therefore, the differences between basic and advanced, which are strongest around 30 km, cannot be due to the line of sight sampling.

P15, l26: how are large disagreements in the north polar region so confidently connected to problems with tropical upwelling? Could this not be an issue with mixing across the polar vortex?

P22, l17: I would avoid the term "mixing levels" as it could be taken as meaning isentropic surfaces.

P22, l19: Is this result based on looking at the best agreement between CMAM and ACE-FTS under the natural variability of the CMAM simulations? Located here within the discussion of the TPL, it comes across a little as one has tuned CMAM, which I think is not the case. Also, "a reduction from the fitting estimate" is unclear to me, is this the best fit of the TPL parameters to the CMAM climatology?

P22, l21,22: First sentence implies best agreement when epsilon is increased–which,

based on previous description I take to mean a mixing rate should be increased. But the following sentence says mixing "times" should be increased, which actually means rates should decrease. Some clarification here would be useful.

P23, l5: Figure 15 is quite dense, and really could use better description in the main text and figure caption. The "level of agreement" between CMAM and ACE-FTS needs to be explained fully, what kind of quantity is this? It took me some time to determine that the white-to-black shading and the white isolines were describing the same quantity. Also, there are bluish boxes which are barely detectable in the plots, are these the "agreement matrices"? As mentioned in the general comments, these agreement boxes don't seem consistent with the ACE-FTS "agreement" scale.

P23, 19: How are these thresholds chosen? 0.65 seems like a rather lenient agreement threshold, since the agreement values shown in Fig 15 go as low as 0.1.

P23, l20: what chemical species is Fig 16 based on?

P24, l32: If mixing and the meridional residual circulation are both driven by Rossby wave breaking, it is hard to see how insufficient mixing could be the cause of a too-rapid BDC.
* * *

---

## Referee Comment (RC3) · Anonymous Referee #3 · 21 Jun 2017

In this paper, the authors have used numerous techniques to compare a nudged CMAM model run with long-lived tracer observations from ACE-FTS. Their sampling technique is excellent and allows for a like-to-like comparison, as much as possible when comparing model output and data. The use of the TLP model as well as the tracer-tracer JPDFs are appropriate. I think this paper is appropriate for publication in ACP, subject to addressing a significant number of concerns about the dynamical interpretation and discussion of the results. I therefore waver between minor and major revisions. Only one of my comments requires additional calculations.

Generally, the following need to be addressed: 1) Improved discussion of BDC 2) Improved discussion of pathways for mixing across the tropopause and if feasible, 3) More specific discussion of the implications of these results for model development

[Figure]

P3L12: Hardiman et al. (2017) found a time of emergence for a trend of 30 years and showed that any trend less than 12 years could be the wrong way due to dynamical variability. The results of Mahieu and pretty much all of our observational records are too short. P3L13: I don't think there are fundamental questions about the mechanisms driving the stratospheric circulation. The mechanisms driving changes to the circulation are less clear, but the fact that data don't show the trend predicted to emerge from models over a much longer timescale is not surprising in light of the results of Hardiman et al. P3L16: Is it true that understanding how the structure of the BDC will change depends greatly on the ability to simulate its current behavior? The models examined by Butchart et al. 2011 have pretty different mean upward mass flux at 70 hPa, but the community still interprets their agreement on the strengthening of the BDC as robust. Getting the mean and present day right are important, but not necessarily for the trends. P3L16: "Typically"–please provide some citations demonstrating how typical.

P4L11: You haven't definced CMAM30 before. P4L13: I would appreciate some discussion either here or in 2.2.2 of the potential problems with CMAM30. In particular, the model is nudged to reanalysis. As far as I am aware, neither the nudging process nor the reanalysis itself conserves mass, energy etc. Are there any studies that show how that influences tracers? Are your tracers transported conservatively and do their budgets close? I'm not necessarily suggesting you calculate the tracer budgets, though such analysis might be interesting, but please address these concerns to whatever extent you can. P4L17: "morphologies" of CFC11, CFC12 and N2O.

P6L19: the model isn't being constrained to follow observations. It's being constrained to the reanalysis, which is a model-data product that is our best guess at a representation of reality.

Section 2.2.4: How does CMAM BDC compare to ERA-I BDC? Mean tropical w* at a few levels would be sufficient. The speculation in this section is not necessary when you can do direct comparisons. Additionally, this section would benefit from considering the extratropical vs. tropical age difference (e.g. Neu and Plumb 1999, Linz et al. 2016)

rather than just talking around the relationship between the age and the circulation. For example, the near-zero differences in 2a between 50S and 50N do not mean that the lower branch of the BDC is the same in the two simulations because the polar age on the same level is older in the free running model. This discussion would be aided by the conversion to isentropic coordinates. This section is one place to address general comment 1) above. E.g. "filtered out close to the tropopause" could be explained in terms of the physical mechanisms of wave propagation (Charney Drazin).

P12L11: "CHAM"→ "CMAM" P12L30: "Air in the polar vortex ... representative of older air ..." The terminology "representative of" is confusing to me. Isn't air in the polar vortex composed of a larger fraction of older air transported from upper levels? P12L34-35: If the variability of the vortex edge is responsible, why is there so much difference in the middle of the vortex, and why is the Southern hemisphere, where the vortex variability is much weaker pretty comparable to the Northern hemisphere?

P13L17: No comma before between.

P14L5-9: I found this section strange. "readily observed" where? The other information seems redundant. Perhaps just remove all together? P14L16-17: Redundant‐rephrase. P14L17-18: "Likely caused by significant differences in the conditions of the influence of downwelling..." This is confusing. Please rephrase or explain further. There is more downwelling in the SH vortex and you've mentioned later that N2O has a source higher up, so shouldn't there be more N2O in the SH vortex than the NH vortex?

P15L11: Another place to address 1). The BDC is strongest in the NH winter because of the climatological westerlies and the enhanced wave driving from the troposphere both. If there were more waves in the NH summer, they wouldn't do any good because they can't propagate up into the stratosphere when there are climatological easterlies. P15L28: not sure what you mean by "robust"

P16L1: "significantly" is confusing. Significant with respect to what? P16L5:

"particular"→"particularly" P16L5-6: Please explain more why these differences would be due to the polar vortex behavior. I agree with you, but a discussion of the mechanism would be helpful.

P17L4: no commas offsetting "with a stratospheric sink" P17L26: no "was" before "passed"

5.2: This discussion needs to be revised. Specifically, please review the literature that treats the tropopause as a "barrier", review the recent literature on transport across the tropopause (Randel 2017 tropospheric dry layers or Randel 2016 asian monsoon transport, for example), discuss the difference between what has been defined here and the more typical treatment of stratospheric intrusions (tropopause folding events that cause deep stratospheric intrusions—see work by Meiyun Lin, for example). Compare to stratospheric intrusion climatology (Skerlak et al. 2014). Finally, please validate your method for defining intrusion events by looking at the colocated water vapor and ozone concentrations in the model. (Or other stratospheric tracers, you could use PV.)

P20L19-20: Which differences and how? I must have missed the discussion previously, so a brief repetition here couldn't hurt. P20L25-6: Have you demonstrated this? If so, how?

P21L1: add "timescale" after transport. P21L7: This needs to be more specific. How were the turn around latitudes determined? Were they monthly mean or instantaneously calculated? P21L18-9: Wording is informal P21L23-5: You've already talked about Fig. 14, so why is this here?

Section 6: More discussion of the TLP would be useful. I am very familiar with it, but Eric's paper is complicated, so a brief discussion of why it's great and useful here would be helpful for the average reader who isn't going to want to read through his whole paper. P22L14: "mixing levels" I thought it was mixing efficiency. P22L17: Be more specific—e.g. "As both the residual circulation and the mixing are driven by

wave breaking, a weaker residual circulation likely correlates with less mixing and thus longer mixing timescales"

P22L25-6: The implication here and elsewhere in this section is that there is some knob to turn to "change" w* or epsilon. There obviously isn't, and so while this paper has diagnosed that the mixing is too weak in JJA, for example, it hasn't come close to determining changes required in the CMAM30HR simulations to match the observations. Certainly changing the language here and P23L14, L29, P24L7 is necessary. If feasible, some discussion of what does set w* (epsilon is probably harder) would be great. If anyone has looked at EP flux divergence esp. broken down by wavenumber, even in CMAM and not necessarily CMAM30, that would be a great thing to discuss here. The authors have done plenty to warrant publication and so they don't need to do it if it hasn't been done. Regardless, some discussion of what does drive the circulation and the mixing in the model would be appropriate here. P24L23: If you've proven that the BDC is too rapid, say so here. As far as I recall, you said that it might be too strong. P24L33-4: Nonsense. Insufficient mixing does not cause any changes to the BDC (at least to first order—second order effects on ozone and the corresponding heating due to ozone might be minor but that has definitely not been addressed in this paper).

---

## Author Comment (AC1) · 30 Oct 2017

*RC = Reviewer Comment*
**AR = Author Response**

*RC: The authors compare free-running and nudged simulations using the CMAM middle-atmosphere model to ACE-FTS measurements of long-lived tracers. The analysis is generally well grounded and based on established analysis techniques, such as tracer-tracer correlation plots. Generally my impression is that the paper takes in a lot of information, making this a fairly dense read. For the future, I recommend to the lead author to break up such works into into smaller, separately publishable pieces. I don't think it would be adequate to recommend this course of action for the present paper as this is only a matter of presentation. The captions of some figures could be more detailed; for examples see below. In the comparison of models versus satellite measurements, a more thorough discussion of the effect of measurement uncertainty on tracer-tracer plots would be desirable. For example, the density plots of $N_2O$ versus CFC-11 and CFC-12 (figures 13a and c), in the case of the satellite, are probably affected by measurement noise giving the JPDFs a fuzzy appearance. Such noise is absent in the model, making for a skewed comparison. Possibly averaging kernels of the ACE-FTS measurements could be used to define random noise to be added to the model data, making them more comparable to the measurements. If this is not practical, at least some text to this effect would be good to have.*

**AR: We thank Referee #1 for their thoughtful comments on our manuscript. We recognize the manuscript is long but feel that to have a complete discussion, the various components of the paper that have been included here are necessary. We will consider this feedback on future manuscripts. As per your suggestions below, some figure captions have been edited. The ACE-FTS retrieval does not routinely produce averaging kernels so these are not available for the analysis. Also, a discussion of the effects of measurement uncertainty has been added to the JPDF discussion (detailed below).**

*RC: Similarly, the discussion of differences in stratosphere-troposphere intrusions / extrusions mentions that resolution might factor into this comparison. At least for the detection of such structures in the data, this can be accounted for by removing small scales from the satellite data using a low-pass filter. Then the two datasets are nominally at the same resolution. This would however not address that the simulation of cut-off systems is fundamentally sensitive to numerical diffusivity in the model, causing reduced incidences of such systems.*

**AR: Philosophically, we have tried to approach the measurement-model comparisons in this study as directly as possible, knowing that the model is always representing a smoother version of reality than ACE-FTS does. We decided to not filter the measurements in any way so that the differences due to small scale features could be identified. If we were to apply a low-pass filter to the observations, we would not be able to answer the questions we posed. In addition to this, we would not be able to apply a three-dimensional low-pass filter on the ACE-FTS profiles (which would be**

**the only appropriate way to do such a comparison since the model fields are effectively smoothed across all three dimensions). The point about the finite resolution of the model is well taken and should be mentioned, but to start to modify the observations seems like a slippery slope.**

**P18L13-14 has been changed to reflect this sentiment:**

**Original sentence: "Understanding how CMAM30HR simulates this exchange assists in the interpretation of mixing effectiveness in the model and the impact of its vertical resolution on the comparisons."**

**New sentence: "Understanding how CMAM30HR simulates this exchange assists in the interpretation of mixing effectiveness in the model and the impact of its finite resolution on the comparisons."**

*RC: Regarding the differences in age-of-air between the free-running and nudged version of the model, my impression is that this is partly caused by mass non-conservation in the nudging fields, whereby artificial divergence caused by relaxation towards reanalyses causes noise in the vertical motion fields. The effect of this might be increased numerical diffusion and a reduced age. Since CMAM is based in a spectral dynamical core, one could consider, in separate experiments, to only nudge divergence or only vorticity, to try to control this behaviour.*

**AR: This is an interesting idea that we will pass along to the CMAM development team.**

*RC: On the whole, the above amounts to a recommendation to publish after a minor revision.*

*Minor comments:*

*RC: P3L3: This sentence reads a little awkwardly – modelling and observations are independent activities. How about "The BDC is well characterized in models but remains poorly constraint in obs" or so?*

**AR: We agree that this sentence should be rephrased.**

**Original sentence: "Despite significant progress in modelling, the BDC has been poorly constrained by observations (Butchart, 2014)."**

**New sentence: "The BDC is well characterized in models but remains poorly constrained by observations (Butchart, 2014)."**

*RC: P7L10: It's certainly possible to rescale the fields to construct approximations for the other tracers. But this requires further assumptions.*

**AR: We chose to use a parallel set of halocarbon species with adjusted boundary conditions because we felt the assumptions that would be required to construct approximations would complicate the interpretations of the measurement-model comparisons. Given the time-varying relative contributions of individual halocarbons to a particular model species in the troposphere, as air parcels enter the stratosphere and air parcels of different ages are mixed together, to untangle the contribution would require assumptions about both the mean age and the full age spectrum. It was felt that such an approach would introduce significant uncertainties.**

**Original sentence**: **"Because of the time-varying contribution of the individual halocarbons to the tropospheric concentration of the model species, it is not possible to re-scale the concentration of the model species to recover a concentration that could be compared with observations."**

**New sentence: "Because of the time-varying contribution of the individual halocarbons to the tropospheric concentration of the model species, the numerous assumptions that would be required to rescale the model species concentration to recover a concentration that could be compared with observations would introduce significant uncertainties."**

*RC: P7L20: Worth mentioning / discussing Meinshausen et al., Geosci. Model Dev., 2017 here. They have constructed boundary conditions for CMIP6 simulations that follow very similar ideas.*

**AR: The following sentence has been added after the sentence on P7L20:**

**"The application of hemispherically-defined lower boundary conditions based on observations is consistent with the proposed approach for the upcoming sixth phase of the Coupled Climate Model Intercomparison (CMIP6) project (e.g. Meinshausen et al., 2017)."**

*RC: P8L6: It remains a little unclear to me how you can have systematic differences between the LBCs used to constrain the simulations and the long-term observations, when the obs were used to construct the LBCs.*

**AR: The halocarbon LBCs were hemispheric averages obtained from the HATS network. While the data from the individual stations contributed to the hemispheric average, there are latitudinal dependencies in the surface measurement data. These comparisons indicate that differences exist but they are very small. The N$_2$O comparisons exhibit larger differences because the LBC was a global average.**

*RC: P8L27ff: Perhaps not drag but noise in the vertical motion. The w fields in the nudged and free-running model might show some differences.*

**AR: We agree that differences in the vertical motion between the free running and nudged simulations could certainly be contributing to the differences in the residual circulation. A figure has been provided below to demonstrate the differences in *w\** in the tropical region. Our aim with including a comparison to the free running simulation was only to provide an important caveat to the results for the CMAM30 nudged simulation. While the use of a nudged simulation allows for a**

time- and space-matched comparison to the ACE observations, we wanted to make the reader aware that the residual circulation in CMAM30, along with age of air and the distribution of long-lived tracers, is different than that which we find in the freely running version of the model. An analysis of the cause of these differences would be a completely separate study and is well outside of the subjects addressed here. We have modified sections of the paper where we speculate on possible causes of the differences to make it clear that the reasons for the differences are unclear at the present moment and are outside of the scope of the paper.

[Figure]

*RC: P14L10: Perhaps insert "annual-mean" and some time information here (which period does the average represent?) Likewise in the caption, here and elsewhere.*

**AR: Annual-mean has been added to this line as well as the Fig.6 caption.**

**Original sentence: "The zonally-averaged distribution of $N_2O$ is presented in Fig. 6.".**
**New sentence: "The zonally-averaged annual-mean distribution of $N_2O$ is presented in Fig. 6.".**

**Original line in Fig 6 caption: "Zonally averaged latitude-altitude distributions of $N_2O$."**
**New line in Fig 6 caption: "Zonally-averaged annual-mean latitude-altitude distributions of $N_2O$."**

*RC: P20L3: Here's where the above comment on model resolution applies. The key difference is not that the two fields are at different resolutions (that could be easily fixed) but that the finite resolution of the model leads to differences in the formation and lifetimes of the cut-off systems.*

AR: We agree with the reviewer that the capability of a relatively low-resolution model such as CMAM (at T47) to correctly model the dynamical evolution of cut-off systems that produce intrusions would be a consideration for a freely running model. In this analysis, however, the dynamical fields are nudged to reanalysis fields derived from a high resolution atmospheric model (T255 for ERA-Interim). Since the synoptic scale and larger (to T21) in CMAM30 are nudged to the ERA-Interim data, there should be, though admittedly it has not been shown, a good representation of the formation and lifetimes of cut-off systems. As for homogenizing the resolution of the observations and model, we note that ACE provides high resolution (up to ~3 km) vertical profiles but only about 15 profiles a day, so these profiles are widely spaced horizontally. While it would be possible to perform vertical smoothing of the observations, it is not clear whether the smoothed observations would be more comparable with the model, as the model fields are the result of a finite horizontal and vertical resolution.

Original sentence: "Therefore, it is unlikely that there is a physical mechanism or deficiencies in the model leading to the differences observed in Fig. 11 and the differences are primarily due to the model resolution."

New sentence: "Therefore, the differences are primarily due to the finite horizontal and vertical resolution of the model, which leads to differences in the representation of stratosphere-troposphere exchange events."

*RC: P34: More detail in the caption please. Which species, which network, which measurement principle, why are there these systematic differences when the measurements had been used in constructing the LBC for the model?*

AR: The boundary conditions for $N_2O$ were not derived in a special manner for the CMAM30 run. The global averages of $N_2O$ were based upon an older IPCC Assessment Report - the A1b scenario for the 4th Assessment Report and also used for CCMVal-2. The $N_2O$ time series uses observations only up to 2000, then it becomes a projection so there are differences to be expected there. The CFC-11 and CFC-12 measurements used in the comparisons in Fig. 1 are an updated version compared to the data that was used for the boundary conditions in the CMAM30 simulations. The minimal differences observed are due to updated values in the observations. Further detail has been added to the caption of Figure 1.

Original Figure 1 caption: "Comparison of CMAM30HR run to surface measurements, relative differences calculated as the site subtracted from the CMAM30HR simulation, divided by the average of the two, as described in the text. The differences and the uncertainties included are the mean and standard deviation of relative differences over the time series."

New Figure 1 caption: "Comparison of CMAM30HR simulations of CFC-11 (blue x), CFC-12 (green diamond), and $N_2O$ (black circle) to the HATS surface flask network of measurements at various locations around the world. Locations of measurement sites are indicated by latitude. Relative differences are calculated as the difference between the concentration at the surface site and the lowest model layer of the nearest neighbor gridbox to the site in the CMAM30HR output,

**divided by the measured concentration. The relative differences were calculated based on the monthly averaged observations and simulations. Shown here are the mean of the differences between May 2004 and June 2010 and the error bars indicate one standard deviation of the mean of the relative differences over the time period."**

*RC: P46: Here's where I think measurement uncertainties make this a skewed comparison. The model output would ideally be folded with the averaging kernels and a-priori assumptions used in the retrievals of the ACE-FTS measurements before comparison with those measurements.*

**AR: We agree that it would be ideal to incorporate averaging kernels and a-priori assumptions in measurement-model comparisons. However, we are unable to do this because there are no averaging kernels available for the ACE-FTS dataset because we do not use optimal estimation in the retrieval process, and so we have no averaging kernels.**

**To address the measurement uncertainty concerns, the following text has been added to the end of section 3.1.2:**

**"Hegglin and Shepherd (2007) have shown the impact of ACE-FTS measurement uncertainties in joint PDFs by comparing the full model output, subsampled model output, and ACE-FTS measurements. They found that there was larger variability in the ACE-FTS joint PDFs compared to those of the subsampled CMAM output."**

---

## Author Comment (AC2) · 30 Oct 2017

*RC = Reviewer Comment*
**AR = Author Response**

*RC: This manuscript describes a detailed set of comparisons between specified dynamics simulations with the CMAM chemistry climate model and satellite-based observations of stratospheric long-lived tracers, with the goal of identifying discrepancies and therefore errors in the simulated stratospheric dynamics. The topic is suitable for ACP, and the conclusions reached by the authors are generally reasonably well supported by the analysis presented.*

**AR: The authors thank Reviewer #2 for their helpful comments on the manuscript. We have addressed both the general and specific comments below.**

*General comments:*
*RC1. I have no doubt that the "advanced" comparison technique–which samples the model data along the actual line of sight of the satellite instrument–is the best way of minimizing sampling errors in the CMAM vs. ACE-FTS comparison. But, the explanations given in Sec. 3.3 don't quite make sense to me. If the difference between the advanced and intermediate methods is so small, as shown in Fig. 4c, this implies that the "line of sight" sampling is actually making very little difference to the sample means. The difference between advanced and basic (fig 4b) is much larger, which means that the most important source of sampling error has to do with a bias in the distribution of samples within each 5deg latitude bin (which is fixed by performing the 2D horizontal interpolation rather than using the closest neighbor gridpoint). It's possible these issues would be easier to sort out if Fig 4 showed differences between the 3 methods and the full model sampling, rather than differences between the advanced sampling and the other sampling methods. Or perhaps BASIC-FULL, INTERMEDIATE-BASIC, and ADVANCED-INTERMEDIATE. In any case, I believe the logic of the explanations here could be sharpened.*

**AR: We chose to represent this comparison as the difference between the Advanced method and the others to illustrate the differences between what would be "observed" by ACE-FTS compared to the full model and common sampling techniques. For a model at T47 resolution, it is true that the intermediate and advanced sampling are very similar. However, for models with finer spatial resolution, the differences would be more apparent. Therefore, we have used the advanced method for ACE-FTS/model comparisons. The text has been clarified to minimize the expectation for a comparison of each method to the full model output.**

**Original sentence: "To determine the impact of sampling the model output at varying levels of detail, three methods were tested and compared to the full model output (the CHAM30HR output at all latitudes and longitudes for each 5 ˚ latitude bin) between June 2004 and May 2010."**

**New sentence: "To determine the impact of sampling the model output at varying levels of detail, three methods were tested by sampling the full model output (the CMAM30HR output at all latitudes and longitudes for each 5° latitude bin) between June 2004 and May 2010."**

*RC2. No doubt there is much more information given in the Ray et al. (2016) paper, but Sec. 6 requires a little more guidance on the set up of the TLP simulations. At some point, in passing, we learn that there were 480 TPL simulations, but it is not said how these simulations differ; presumably certain input assumptions are varied in the different simulations, but not, it seems, w\* and epsilon directly. Also, the terms w\* and epsilon should be better defined. At some point, w\* is introduced as a mean tropical upwelling, but it is later used to quantify vertical motion in the extratropics. It's also not really clear if epsilon is a prognostic or diagnostic variable, and how it depends on height, latitude, time, etc.*

**AR: To maximize clarity and minimize additional text, the second paragraph of section 2.3 has been moved to the beginning of section 6 and the first sentence of section 6 has been moved to the end of section 2.3.**

**The sentence moved to section 2.3 from the first sentence of section 6 has been changed to: "The TLP model is used here to identify the changes to the CMAM30HR tropical upwelling and effective mixing that may improve the comparisons between ACE-FTS and CMAM30HR. However, this tool does not identify a specific mechanism but it can isolate seasons and regions in which changes are required."**

**The following sentences were added to the end of the paragraph moved (now the first paragraph of section 6):**
**"There were 480 simulations initialized with different combinations of *w\** (velocity of tropical upwelling) and $\varepsilon$ (the mixing efficiency) settings. The fraction of the CMAM30 *w\** used to initialize the TLP model ranged from 0.20 to 1.24 and the $\varepsilon$ ranged from 0.18 to 1.50. In each TLP simulation run, the relationship between mean circulation and mixing is constrained by the vertically-averaged mixing efficiency [Ray et al., 2016]. The mixing efficiency in the TLP model is defined as $\varepsilon = \alpha / \lambda\tau$, where $\alpha$ is the ratio of tropical to extratropical mass, $\lambda$ is the rate of the mean circulation influence, specifically the mass flux out of the tropics due to mean circulation, and $\tau$ is the mixing time or time scale for mass flux between the tropics and extratropics [Ray et al., 2014; Garny et al., 2014]."**

*RC3. I have trouble following the logic from Figure 15 to 16. Figure 15 seems to show that best agreement with the ACE-FTS measurements is achieved running the TLP model with parameter settings which produce the smallest w\* and largest epsilon values (at least in Fig 15a,b,c). In fact, it seems that the range of TLP simulations is not large enough to find the actual best agreement with ACE-FTS–a point which could be discussed. But then, in Figure 16, it is implied that e.g., best agreement with ACE-FTS is produced with no significant change in w\* values in the tropics. Something seems inconsistent here.*

**AR: To clarify the link between changes in CMAM that may improve the comparisons to ACE-FTS, we have updated Fig. 16 to include the individual contributions of both CFC-11 and CFC-12. The thresholds used for all latitude ranges in the new Fig. 16 have also be reduced to 0.2 (as was used for the tropics) to make the regional comparisons more consistent. Deciding to use a more**

**restricted but consistent threshold leads to no result in some seasons for some geographic regions, perhaps suggesting that the range of *w\** and *ε* values should be extended.**

*RC4. For many of the difference contour plots, it would be helpful if the colorbars were chosen such that positive differences could be more easily differentiated from negative differences. An example is Fig 6, where it is very difficult to know whether the CMAMACE differences in the UTLS are positive (like the middle stratosphere differences) or negative.*

**AR: The colorbars of the difference contour plots have all been changed to a red/white/blue scheme where white is zero.**

*Specific comments*

*RC P1, l11: "The model consistently: : :" could be taken out of context–this conclusion is specific to the trace gases examined in this study (and probably wouldn't apply to ozone, for example).*

**AR: We have clarified this statement.**

**Original sentence: "The model consistently overpredicts tracer concentrations in the lower stratosphere, particularly in the Northern Hemisphere winter and spring seasons."**

**New sentence: "The model consistently overpredicts tracer concentrations of CFC-11, CFC-12, and $N_2O$ in the lower stratosphere …"**

*RC P1, l14: the "too little isentropic mixing" should probably be connected if possible to a height or range of heights.*

**AR: We have clarified this statement.**

**Original sentence: "In particular, the CMAM30 simulation exhibits too little isentropic mixing in the June-July-August season."**

**New sentence: "In particular, the CMAM30 simulation exhibits too little isentropic mixing in the tropical lower stratosphere during the June-July-August season."**

*RC P2, l1: I'm not sure this is the only reason for the increase in interest in stratospheric transport–and it's a bit of a chicken and egg problem.*

**AR: We have included a reference to Butchart (2014) to support this statement.**

**Original sentence: "Interest in stratospheric transport has increased over the last 20 years as a result of significant developments in stratosphere-resolving general circulation models (GCMs) and chemistry-climate models (CCMs) (e.g., Pawson et al., 2000; Eyring et al., 2005; SPARC-CCMVal, 2010; Gerber, 2012)."**

**New sentence: "As highlighted by Butchart (2014), interest in stratospheric transport has increased over the last 20 years as a result of significant developments in stratosphere-resolving general circulation models GCMs (e.g. Pawson et al., 2000; Gerber, 2012) and chemistry-climate models (CCMs) (e.g. Eyring et al., 2005; SPARC-CCMVal, 2010)."**

*RC P2, l4: \*Accurate\* projections rely on good models.*

**AR: Accurate has been added.**

**Original sentence: "Projections of stratospheric ozone and climate rely on the ability of these models to simulate stratospheric transport and chemistry."**

**New sentence: "Accurate projections of stratospheric ozone and climate rely on the ability of these models to simulate stratospheric transport and chemistry."**

*RC P2, l5: Definitely the distribution of long-lived trace gases depends on the BDC... for short-lived species it may not have that much influence.*

**AR: The term "long-lived" has been added to the sentence.**

**Original sentence: "It is clear that the transport of chemical tracers will be impacted by changes in the BDC, which will in turn influence ozone recovery projections, lifetimes of ozone depleting gases, and mass exchange between the troposphere and stratosphere (Butchart, 2014)."**

**New sentence: "It is clear that the transport of long-lived chemical tracers will be impacted by changes in the BDC, which will in turn influence ozone recovery projections, lifetimes of ozone depleting gases, and mass exchange between the troposphere and stratosphere (Butchart, 2014)."**

*RC P3, l14: "has" or "have"? The word choice depends on whether the models project changes in the BDC (plural) or evidence of those changes (singular).*

**AR: Based upon the feedback of Reviewer #3, this sentence has been changed.**

**Original sentence: "Fundamental questions remain as to the mechanisms driving the stratospheric circulation because there has been evidence of changes in the BDC that has not been projected by models (e.g., Butchart, 2014; Mahieu et al., 2014)."**

**New sentence : "Discrepancies between observations and model projects may be due to the short time scales of observation systems (Butchart, 2014; Hardiman et al., 2017)."**

*RC P4, l31: ACE-FTS measurements have high vertical resolution, not the instrument itself.*

**AR: That is correct, the ACE-FTS retrieved profiles have a high vertical resolution. The sentence has been clarified.**

**Original sentence: "ACE-FTS is ideal for studying 30 the vertical structure of constituent gases from cloud tops to 100 km; it is particularly useful in the upper troposphere and lower stratosphere because of its high vertical resolution (Hegglin et al., 2008)."**

**New sentence : "ACE-FTS is ideal for studying the vertical structure of constituent gases from cloud tops to 100 km; the retrieved profiles are particularly useful in the upper troposphere and lower stratosphere where the vertical resolution is approximately 3 km  (Hegglin et al., 2008)."**

*RC P8, l2: For comparisons of model to measurements, it's probably more intuitive to treat the measurements as the truth, and show relative differences of the model with respect to the measurements, rather than the mean of the measurements and model. It's of course not a big deal as long as it is clear how the calculation is being done, but I feel the simpler the calculation, the easier it is to interpret.*

**AR: The comparisons between measurements and model output of tracer concentrations throughout the paper used the same method of calculating comparisons with respect to the mean of the measurement and model. The authors agree that the comparison would be more intuitive with respect to the measurements. Therefore, all relative difference calculations have been updated throughout the paper and are now relative to the measurement.**

*RC P8, l8: this statement of significance applies to the 1 sigma confidence level. For 2 sigma, I guess all differences would be not significantly different from zero.*

**AR: True, we have clarified the sentence to reflect this detail.**

**Original sentence: "Over the time period compared, CMAM30HR appears to overpredict CFC-11 at all HATS sites while the CFC-12 comparisons are not significantly different from zero for all but two sites in the Southern Hemisphere."**

**New sentence : "Over the time period compared, CMAM30HR appears to overpredict CFC-11 at all HATS sites while the CFC-12 comparisons are not significantly different from zero within one standard deviation for all but two sites in the Southern Hemisphere"**

*RC P9, l24-26: Apparent contradiction between "increased isentropic mixing" and "slower shallow branch".*

**AR:  You are correct. There is a problem with this description.  Relatively old mean age in the lowermost stratosphere is due to either a relatively strong upper branch of the BDC compared to the lower branch, and/or less isentropic mixing since that process brings young tropospheric air**

into the lowermost stratosphere. In the LMS, increased mixing brings in young air because it's primarily tropospheric.

Original sentences: "... CMAM30 clearly has older air in the extra-tropical lowermost stratosphere. It is potentially caused by either stronger downwelling of the older air from above, consistent with a stronger BDC, or increased isentropic mixing of tropospheric air from lower latitudes (e.g. Hegglin and Shepherd, 2007). Therefore, the differences in age appear to suggest a slower shallow branch or a faster deep branch of the BDC."

New sentences: "... CMAM30 clearly has older air in the extra-tropical lowermost stratosphere. It is potentially caused by either stronger downwelling of the older air from above, consistent with a stronger BDC, or reduced isentropic mixing of tropospheric air from lower latitudes (e.g., Hegglin and Shepherd, 2007). Therefore, the differences in age appear to suggest a slower shallow branch or a faster deep branch of the BDC."

*RC P12, l19: the end of this sentence could be misconstrued, as of course there has been vertical interpolation applied in the translation to the vertical levels of the ACEFTS retrievals. I would suggest to remove this last part, and write ": : : location of the tangent points with altitude".*

AR: We have made the change suggested.

Original sentence: "... no consideration of the variation in geographical location of the tangent points ..."

New sentence: "... no consideration of the variation in geographical location of the tangent points above or below 30 km."

*RC P13, l25: But the INTERMEDIATE sampling technique is also limited to the 30 km tangent altitude, and the differences to the advanced method are much smaller. Indeed, by construction, differences between a method using the variable tangent heights and one using only the 30 km tangent height should be zero at 30 km (which appears to be the case in the advanced-intermediate comparison). Therefore, the differences between basic and advanced, which are strongest around 30 km, cannot be due to the line of sight sampling.*

AR: The basic sampling uses a 30 km tangent height but the nearest model column and the intermediate sampling produces a profile by computed the bilinear interpolation of the concentrations of the nearest 4 grid boxes at each point along the column. It is the intermediate sampling that we should expect to have a zero difference with the advanced sampling at 30 km. This is the case in Fig. 4c.

*RC P15, l26: how are large disagreements in the north polar region so confidently connected to problems with tropical upwelling? Could this not be an issue with mixing across the polar vortex?*

AR: The sentence has been changed.

**Original sentence:** "The large disagreements in the north polar region during winter and spring indicate that the upwelling portion of the BDC across the different seasons is not well characterized by CMAM30HR."

**New sentence :** "The large disagreements in the north polar region during winter and spring indicate that the downwelling portion of the BDC across the different seasons is not well characterized by CMAM30HR."

*RC P22, l17: I would avoid the term "mixing levels" as it could be taken as meaning isentropic surfaces.*

**AR: This has been changed to mixing efficiency throughout the manuscript, which is more accurate.**

*RC P22, l19: Is this result based on looking at the best agreement between CMAM and ACE-FTS under the natural variability of the CMAM simulations? Located here within the discussion of the TPL, it comes across a little as one has tuned CMAM, which I think is not the case. Also, "a reduction from the fitting estimate" is unclear to me, is this the best fit of the TPL parameters to the CMAM climatology?*

**AR: The TLP model was tuned to the CMAM30 output before it was used to test a range of $w*$ and $\varepsilon$ values. Line 16 of page 22 states "... the TLP model was tuned to be representative of CMAM30HR by fitting estimates of the mean tropical upwelling ($w*$) and mean mixing levels ($\varepsilon$) …".  The reduction from the fitting estimate refers to the change required in the CMAM30HR run based on the tuned TLP model.**

*RC P22, l21,22: First sentence implies best agreement when epsilon is increased–which, based on previous description I take to mean a mixing rate should be increased. But the following sentence says mixing "times" should be increased, which actually means rates should decrease. Some clarification here would be useful.*

**AR:  We have decided to revise the paragraph to clarify our meaning.**

**Original paragraph 2 of section 6:**
**"A reduced mean circulation would likely correspond to less mixing and longer mixing times. Mixing levels are defined in the TLP by the ratio of horizontal mixing mass flux to horizontal mean mass flux scaled by the width of the tropical pipe region (Garny et al., 2014). Ray et al. (2016) found that the CMAM30HR simulations best match the ACE-FTS measurements when the $w*$ is between 0.27 mm/s and 0.32 mm/s (a reduction from the fitting estimate of 0.4 mm/s) and ε ranges from 0.7 to 1.2 (an increase from the fitting estimate of 0.55). Based on the in-mixing time profiles shown by Ray et al. (2016), it is apparent that the CMAM30HR mixing times need to increase to slow down the mixing at all levels, although the differences are only significant in the lower part of the stratosphere, below 20 km, and above 24 km. These are physically consistent changes since the mean circulation is driven by wave breaking, which also causes mixing between the tropics and extratropics."**

**New paragraph 2 of section 6:**
**"Ray et al. (2016) found that the CMAM30HR simulations best match the ACE-FTS measurements when the w\* is between 0.27 mm/s and 0.32 mm/s (a reduction from the fitting estimate of 0.4 mm/s) and ε ranges from 0.7 to 1.2 (an increase from the fitting estimate of 0.55). Ray et al (2016) found that since ε is inversely proportional to both λ and τ, there is a compensating effect with changes in w\* (λ) or τ. For the CMAM30HR changes derived, w\* needed to be slowed down significantly below 20 km, and above 24 km.  For constant ε that would result in larger τ (less mixing).  However, Ray et al (2016) found that ε also needed to be increased so there needed to be more mixing than would result from slower w\* and constant τ, but not enough of an increase in ε so that the mixing times were less (more mixing) than CMAM30HR has currently. With the increase in ε, mixing times are reduced but still longer than the current CMAM30HR mixing times."**

*RC P23, l5: Figure 15 is quite dense, and really could use better description in the main text and figure caption. The "level of agreement" between CMAM and ACE-FTS needs to be explained fully, what kind of quantity is this? It took me some time to determine that the white-to-black shading and the white isolines were describing the same quantity. Also, there are bluish boxes which are barely detectable in the plots, are these the "agreement matrices"? As mentioned in the general comments, these agreement boxes don't seem consistent with the ACE-FTS "agreement" scale.*

**AR: We have clarified the text to make it clear that the white isolines are the same quantity as the grey color contours but we are not sure what the blueish boxes the reviewer is referring to. We do recall that on printing the paper sometimes, weird blueish boxes appeared on these plots.  Please refer to the electronic version of the paper and check to see if the blue boxes are there as well.**

**Original sentence:  "The white contours illustrate the agreement isolines."**

**New sentence: "The white-to-black shading is reinforced by white contours of the same quantity to illustrate the comparison more clearly."**

*RC P23, 19: How are these thresholds chosen? 0.65 seems like a rather lenient agreement threshold, since the agreement values shown in Fig 15 go as low as 0.1.*

**AR: As noted above, the thresholds have been changed to be 0.2 for both CFC-11 and CFC-12 in all regions (tropics and extratropics). This is reflected in the updated Figure 16.**

*RC P23, l20: what chemical species is Fig 16 based on?*

**AR: CFC-11 and CFC-12 were both used in Fig 16.  Figure 16 has been changed, as previously noted.**

*RC P24, l32: If mixing and the meridional residual circulation are both driven by Rossby wave breaking, it is hard to see how insufficient mixing could be the cause of a too-rapid BDC.*

**AR: Our results show that the CMAM30 circulation needs to be slower but the mixing efficiency, $\varepsilon$, needs to increase. As explained in response to previous comments, increasing $\varepsilon$ does not necessarily mean increasing mixing.**

---

## Author Comment (AC3) · 30 Oct 2017

*RC = Reviewer Comment*
**AR = Author Response**

*RC: In this paper, the authors have used numerous techniques to compare a nudged CMAM model run with long-lived tracer observations from ACE-FTS. Their sampling technique is excellent and allows for a like-to-like comparison, as much as possible when comparing model output and data. The use of the TLP model as well as the tracer-tracer JPDFs are appropriate. I think this paper is appropriate for publication in ACP, subject to addressing a significant number of concerns about the dynamical interpretation and discussion of the results. I therefore waver between minor and major revisions. Only one of my comments requires additional calculations.*
*Generally, the following need to be addressed: 1) Improved discussion of BDC 2) Improved discussion of pathways for mixing across the tropopause and if feasible, 3) More specific discussion of the implications of these results for model development*

**AR:  We thank Referee #3 for their detailed and constructive comments on our manuscript.  We have addressed the general and the specific comments below.**

*RC P3L12: Hardiman et al. (2017) found a time of emergence for a trend of 30 years and showed that any trend less than 12 years could be the wrong way due to dynamical variability. The results of Mahieu and pretty much all of our observational records are too short.*

**AR: Thank you for identifying this recent publication. The authors were not aware of it.  We have added the following sentence to the end of the paragraph.**

**New sentence: "Recently, Hardiman et al. (2017) determined that a period of 30 years is required for a trend to be identified from noise due to natural variability. They also found that dynamic variability can obscure a trend in the BDC if it is based on less than 12 years of data."**

*RC P3L13: I don't think there are fundamental questions about the mechanisms driving the stratospheric circulation. The mechanisms driving changes to the circulation are less clear, but the fact that data don't show the trend predicted to emerge from models over a much longer timescale is not surprising in light of the results of Hardiman et al.*

**AR: Agreed. Fluid dynamics provides a robust explanation for the mechanism driving the stratospheric circulation. We had meant to refer to the ability of models to quantitatively simulate the contribution of different processes to the BDC and the stratospheric age of air. Perhaps these are not fundamental questions. We have rephrased the sentence.  This was also noted in our response to Reviewer #2.**

**Original sentence: "Fundamental questions remain as to the mechanisms driving the stratospheric circulation because there has been evidence of changes in the BDC that has not been projected by models (e.g., Butchart, 2014; Mahieu et al., 2014)."**

**New sentence: "Discrepancies between observations and model projects may be due to the short time scales of observation systems (Butchart, 2014; Hardiman et al., 2017)."**

*RC P3L16: Is it true that understanding how the structure of the BDC will change depends greatly on the ability to simulate its current behavior? The models examined by Butchart et al. 2011 have pretty different mean upward mass flux at 70 hPa, but the community still interprets their agreement on the strengthening of the BDC as robust. Getting the mean and present day right are important, but not necessarily for the trends.*

**AR: While all models agree on the sign of the change in the BDC, both the absolute and relative rates of change in the BDC show considerable variation across models. In addition to this, while we have confidence in a future acceleration, the authors do not think there is a consensus on the magnitude of the change or how it would be distributed between the shallow and deeper branches of the BDC. It should also be noted that our confidence in a future acceleration is not purely because of the model consensus, but also significantly because of physical processes that are directly tied to changes in the large-scale temperature structure of the atmosphere for which we do have great confidence. We do think that increasing confidence in our future projections of change would be strengthened by having more confidence in our capacity to model the present-day state of the BDC.**

*RC P3L16: "Typically"–please provide some citations demonstrating how typical.*

**AR: We have added four examples to demonstrate this.**

**Original sentence: "This is typically assessed by investigating how capable a model is at simulating tracer concentrations; in particular …"**

**New sentence: "This is typically assessed by investigating how capable a model is at simulating tracer concentrations (e.g. Jin et al., 2005; Allen et al., 2009; Park et al., 2013; Pendlebury et al., 2015); in particular …"**

*RC P4L11: You haven't definced CMAM30 before.*

**AR: CMAM30 is defined on P3L23.**

*RC P4L13: I would appreciate some discussion either here or in 2.2.2 of the potential problems with CMAM30. In particular, the model is nudged to reanalysis. As far as I am aware, neither the nudging process nor the reanalysis itself conserves mass, energy etc. Are there any studies that show how that influences tracers? Are your tracers transported conservatively and do their budgets close? I'm not*

*necessarily suggesting you calculate the tracer budgets, though such analysis might be interesting, but please address these concerns to whatever extent you can.*

**AR: We are not aware of any studies that have investigated the effects of nudging on tracer conservation. While the process of nudging the dynamical fields most definitely violates conservation of energy, we believe that tracer advection is globally conservative as it is in the free-running model. The nudging is applied by relaxation with a 24-hour time constant and will be 'felt' by the model dynamical fields as an additional tendency, similar to that produced from any other parameterization of unresolved physical processes. Therefore we do not believe nudging itself introduces new problems for the conservation of advected tracers. The tracers are advected using spectral advection, which, while not positive definite does conserve global mass. The correction of negative concentrations by 'hole filling' will violate mass conservation and any addition of mass to correct for negatives when spectral fields are transformed to grid-point space is tracked and corrected for. The tracers analysed here, CFC-11, CFC-12 and $N_2O$, are long-lived with 'smooth' distributions that are well represented in spectral space resulting in very little problem with the generation of negatives. No nudging of surface pressure was used and we rely on the standard running correction of the global average surface pressure, the first spectral coefficient of the surface pressure, towards a pre-defined constant to ensure that the background mass of the atmosphere does not exhibit a trend at all.**

**In Section 2.2.2, P7L2 we have added the following text:**
**"As noted, tracers in CMAM30 evolve freely subject to advection by the resolved circulation and vertical redistribution by physical parameterizations. Advection of tracers is calculated using spectral advection, which is inherently mass conservative though not necessarily positive definite. The generation of negative concentrations upon transformation from spectral to physical space is corrected through 'hole filling' with any artificially added mass to remove negatives tracked and corrected for in the global average. The tracers analysed here are long lived and smoothly varying, resulting in spatial distributions that are well represented in spectral space and produce minimal problems with the generation of negative concentrations. No nudging of surface pressure is performed and the global average surface pressure is continually corrected back to a predefined constant value in the CMAM30 simulation, in the exact same manner as is done in free-running simulations. While mass conservation in the CMAM30 simulation has not been analysed specifically, no significant differences with free-running simulations have been seen for quantities such as the evolution of total stratospheric chlorine."**

*RC P4L17: "morphologies" of CFC11, CFC12 and N2O.*

**AR: Thanks for catching this.**

**Original sentence: "Section 4 examines the measured and simulated zonally-averaged morphologies."**

**New sentence: "Section 4 examines the measured and simulated zonally-averaged morphologies of CFC-11, CFC-12, and $N_2O$."**

*RC P6L19: the model isn't being constrained to follow observations. It's being constrained to the reanalysis, which is a model-data product that is our best guess at a representation of reality.*

**AR: You are correct. We have clarified this in the text.**

**Original sentence: "The ability to constrain the dynamical fields to follow the observations more closely enables direct model-measurement comparisons of chemical tracers in the model by eliminating the internal variability in the simulated circulation."**

**New sentence: "The ability to constrain the dynamical fields to follow the reanalysis (the best approximation of reality) enables direct model-measurement comparisons of chemical tracers …"**

*RC Section 2.2.4: How does CMAM BDC compare to ERA-I BDC? Mean tropical w\* at a few levels would be sufficient. The speculation in this section is not necessary when you can do direct comparisons. Additionally, this section would benefit from considering the extratropical vs. tropical age difference (e.g. Neu and Plumb 1999, Linz et al. 2016) rather than just talking around the relationship between the age and the circulation. For example, the near-zero differences in 2a between 50S and 50N do not mean that the lower branch of the BDC is the same in the two simulations because the polar age on the same level is older in the free running model. This discussion would be aided by the conversion to isentropic coordinates. This section is one place to address general comment 1) above. E.g. "filtered out close to the tropopause" could be explained in terms of the physical mechanisms of wave propagation (Charney Drazin).*

**AR: To the authors knowledge, the CMAM BDC has not been compared to the ERA-Interim BDC. While the analysis suggested here may not be well beyond the scope of the paper, the authors feel that including such an analysis would add a lot of text and perhaps additional figures to an already extensive manuscript. As explained in response to a Reviewer #1, the purpose of including a comparison to the free running simulation was only to provide an important caveat to the results for the CMAM30 nudged simulation. While the use of a nudged simulation allows for a time- and space-matched comparison to the ACE observations, we wanted to make the reader aware that the residual circulation in CMAM30, along with age of air and the distribution of long-lived tracers, is different than that which we find in the freely running version of the model. An analysis of the cause of these differences would constitute a separate study. We have modified sections of the paper where we speculate on possible causes of the differences to make it clear that the reasons for the differences are unclear at the present time and are outside of the scope of the paper.**

*RC P12L11: "CHAM" → "CMAM"*

**AR: Fixed. Thanks!**

*RC P12L30: "Air in the polar vortex … representative of older air …" The terminology "representative of" is confusing to me. Isn't air in the polar vortex composed of a larger fraction of older air transported from upper levels?*

**AR: We agree that the use of "representative of" is a bit confusing.**

**Original sentence: "Air in the polar vortex is typically representative of older air brought down from higher altitudes."**

**New sentence: "Air in the polar vortex is typically composed of older air brought down from higher altitudes."**

*RC P12L34-35: If the variability of the vortex edge is responsible, why is there so much difference in the middle of the vortex, and why is the Southern hemisphere, where the vortex variability is much weaker pretty comparable to the Northern hemisphere?*

**AR: This is a valid point, the differences observed are not just due to the variability of the vortex edge but also reflect that ACE-FTS samples the large scale downwelling of air within the vortex differently from the zonal mean average. The text has been modified to reflect this detail.**

**Original sentence: "The differences seen in Fig. 4a occur because comparing the full output to measurements does not account for the variability of the vortex edge in both longitude and latitude."**

**New sentence: "The differences seen in Fig. 4a occur because comparing the full output to measurement-like samples of the output does not account for the variability of the vortex edge in both longitude and latitude, nor does it account for the differences in spatially- and temporally-sampled large-scale downwelling of air within the vortex compared to a zonal mean that includes the model simulation at all longitudes and time periods."**

*RC P13L17: No comma before between.*

**AR: This has been fixed.**

*RC P14L5-9: I found this section strange. "readily observed" where? The other information seems redundant. Perhaps just remove all together?*

**AR: This section on P14 L5-8 has been removed.**

**Text removed: "Most long-lived tracers with tropospheric sources exhibit quantitatively similar behaviour in the upper troposphere and lower stratosphere. In the context of a zonally-averaged tracer morphology, the equator-to-pole gradients of tracer isopleths that are created by the diabatic circulation in the stratosphere are readily observed. By choosing to sample the CMAM30HR**

**output as described above, the behaviour of N$_2$O, CFC-12, and CFC-11 can be investigated thoroughly."**

*RC P14L16-17: Redundant … rephrase.*

**AR: We agree that the sentence could be clearer. It has be edited.**

**Original sentence: "The distributions show a decrease in concentration of N$_2$O with altitude at all latitudes, and also moving from the equator poleward at each pressure level and in each hemisphere.**

**New sentence: "The southern extratropical and Antarctic concentrations of N$_2$O tend to decrease with altitude more rapidly than those in the Northern Hemisphere."**

*RC P14L17-18: "Likely caused by significant differences in the conditions of the influence of downwelling…" This is confusing. Please rephrase or explain further. There is more downwelling in the SH vortex and you've mentioned later that N2O has a source higher up, so shouldn't there be more N2O in the SH vortex than the NH vortex?*

**AR: This sentence has been edited to provide more clarity. The authors were referring to the reflection of the differences in downwelling in the two hemispheres. The source in the upper stratosphere doesn't significantly impact depletion of N$_2$O in older air and there isn't necessarily a hemispheric difference in this source. Therefore, the hemispheric asymmetry in the differences between the observations and simulations is primarily driven by the differences in downwelling and the related differences in the isolation of the vortex.**

**Original sentence: "This is likely caused by significant differences in the conditions of the influence of downwelling within the polar vortex between the two hemispheres."**

**New sentence: "This asymmetry is likely driven by differences in the isolation of the polar vortex in each hemisphere and the large-scale downwelling that is largely dependent on this isolation."**

*RC P15L11: Another place to address 1). The BDC is strongest in the NH winter because of the climatological westerlies and the enhanced wave driving from the troposphere both. If there were more waves in the NH summer, they wouldn't do any good because they can't propagate up into the stratosphere when there are climatological easterlies.*

**AR: We have clarified the sentence at P15L11-12.**

**Original sentence: "In general, the BDC is strongest in the Northern Hemisphere winter due to wave driven enhancements initiated by topography (e.g., Rosenlof, 1995; Plumb and Eluszkiewicz, 1999). "**

**New sentence: "In general, the BDC is strongest in the Northern Hemisphere winter because of wave driven enhancements initiated by topography, and because, during that time of year, climatological westerlies facilitate wave propagation into the stratosphere (e.g., Rosenlof, 1995; Plumb and Eluszkiewicz, 1999)."**

*RC P15L28: not sure what you mean by "robust"*

**AR: The sentence has been changed to clarify the meaning.**

**Original sentences: "However, the shifting of the agreement in the tropical region through the seasons indicates a robust simulation."**

**New sentences: "Meanwhile, the shifting of the agreement in the tropical region through the seasons indicates that the simulation is consistent with the spatial distribution of the observations in this region."**

*RC P16L1: "significantly" is confusing. Significant with respect to what?*

**AR: The comparisons are different in each polar region. The text has been edited to provide more clarity.**

**Original sentence: "The polar regions of each hemisphere in the comparisons of Fig. 8 are significantly different."**

**New sentence: "In the comparisons shown in Fig. 8, the Northern polar region measurement-model differences are significantly different compared to those in the Southern polar region."**

*RC P16L5: "particular" → "particularly"*

**AR: This has been fixed.**

*RC P16L5-6: Please explain more why these differences would be due to the polar vortex behavior. I agree with you, but a discussion of the mechanism would be helpful.*

**AR: The last sentence of this paragraph has been modified to clarify the cause of these differences.**

**Original sentence: "These differences are likely due to the behaviour of the polar vortex in each hemisphere.**

**New sentence: "These differences are likely due to the behaviour of the polar vortex in each hemisphere; in particular, they may be related to the models' (either CMAM30HR or the ERA-Interim model used for the nudging or both) ability to represent transport processes in the strong, cold, quiescent Antarctic vortex versus the warmer and more variable Arctic vortex."**

*RC P17L4: no commas offsetting "with a stratospheric sink"*

**AR: This has been fixed.**

*RC P17L26: no "was" before "passed"*

**AR: This has been removed.**

*RC 5.2: This discussion needs to be revised. Specifically, please review the literature that treats the tropopause as a "barrier", review the recent literature on transport across the tropopause (Randel 2017 tropospheric dry layers or Randel 2016 asian monsoon transport, for example), discuss the difference between what has been defined here and the more typical treatment of stratospheric intrusions (tropopause folding events that cause deep stratospheric intrusionsâ˘Aˇ Tsee work by Meiyun Lin, for example). Compare to stratospheric intrusion climatology (Skerlak et al. 2014). Finally, please validate your method for defining intrusion events by looking at the colocated water vapor and ozone concentrations in the model. (Or other stratospheric tracers, you could use PV.)*

**AR: We have decided to continue to use the term intrusion in this context. The calculation presented in the manuscript is intended to provide a diagnostic for comparison of the observations and the simulations. While water vapor, ozone or other stratospheric tracers could illustrate what is happening in the atmosphere, using the algorithm employed here on other species would not necessarily validate the method. CFC-11 was chosen because of its distinctly different stratospheric loss rates. The algorithm would have to be adjusted for water vapor or ozone. In this work, we are demonstrating that CFC-11 is species that can provide another diagnostic to identify the origin of air similar to water vapour or ozone. In addition to this, utilizing the ozone and water vapour ACE-FTS products would require extensive comparison to the CMAM30 simulations as was done for the halocarbon simulation. The authors have decided to not perform this additional work as it would be a significant undertaking and that is not within the scope of this project. However, we have added a sentence after the first sentence in paragraph 2 of this section to clarify the definition of intrusion used in this work.**

**New sentence: "The diagnostic developed for this analysis is the frequency of intrusions, which signifies the frequency of stratospheric (tropospheric) air penetrating into the troposphere (stratosphere)."**

*RC P20L19-20: Which differences and how? I must have missed the discussion previously, so a brief repetition here couldn't hurt.*

**AR: This sentence has be changed to clarify this point.**

**Original sentence: "This difference between the measurements and simulations is likely due to the overly rapid BDC in the model simulation, as previously discussed."**

**New sentence: "These differences between the measurements and simulations are likely due to the overly rapid BDC in the model simulation, leading to higher concentrations of the trace gases in the simulation, which is consistent with the zonal mean comparisons discussed in Section 4."**

*RC P20L25-6: Have you demonstrated this? If so, how?*

**AR: While we have not demonstrated this explicitly in this manuscript, this was demonstrated by Hegglin and Shepherd (2007). This reference has been added to this sentence.**

**Original sentence: "It is the atmospheric variability that contributes to the variability observed in the ACE-FTS JPDF around 150-200 ppbv of $N_2O$."**

**New sentence: "It is the atmospheric variability that contributes to the variability observed in the ACE-FTS JPDF around 150-200 ppbv of $N_2O$ (Hegglin and Shepherd, 2007)."**

*RC P21L1: add "timescale" after transport.*

**AR: "Timescale" has been added.**

**Original sentence: "This relationship is observed because the photochemical lifetime of CFC-11 is shorter than the time scale for mixing by horizontal eddy transport …"**

**New sentence: "This relationship is observed because the photochemical lifetime of CFC-11 is shorter than the time scale for mixing by horizontal eddy transport timescale …"**

*RC P21L7: This needs to be more specific. How were the turn around latitudes determined? Were they monthly mean or instantaneously calculated?*

**AR: They were determined from the monthly mean vertical velocities. We have added "monthly mean" to sentence.**

**Original sentence: "The data were selected from the tropical latitude region using estimates of the turn-around-latitude, the height-dependent latitude where the tropical upwelling is zero, determined from CMAM30HR vertical velocities."**

**New sentence: "The data were selected from the tropical latitude region using estimates of the turn-around-latitude, the height-dependent latitude where the tropical upwelling is zero, determined from CMAM30HR monthly-mean vertical velocities."**

*RC P21L18-9: Wording is informal*

**AR: The language has been formalized.**

**Original sentence: "However, the differences between the measurements and simulations are primarily in the steepness of this segment of the JPDF, which is a sign of having not enough mixing into the tropics, rather than a product of a too rapid tropical ascent."**

**New sentence: "However, the differences between the measurements and simulations are primarily in the steepness of this segment of the JPDF, which is an indication of insufficient mixing into the tropics, rather than too rapid tropical ascent."**

*P21L23-5: You've already talked about Fig. 14, so why is this here?*

**AR: This has been deleted here and we have added a reference to Fig. 14 in next sentence**.

**Original sentences: "Figure 14 isolates the $N_2O$/CFC-11 JPDFs to the tropical region only, as defined by the turn-around-latitudes, for both the ACE-FTS measurements (left column) and the CMAM30HR simulations (right column) for each season. There is an evolution of the characteristics of the JPDFs shown in this figure."**

**New sentence: "There is an evolution of the characteristics of the JPDFs shown in Fig. 14."**

*RC Section 6: More discussion of the TLP would be useful. I am very familiar with it, but Eric's paper is complicated, so a brief discussion of why it's great and useful here would be helpful for the average reader who isn't going to want to read through his whole paper.*

**AR: As noted in response to Reviewer #2's comments, more discussion of the TLP model has been added and a portion of Section 2.3 was moved to Section 6. The authors think the descriptions are now sufficient for the average reader.**

*RC P22L14: "mixing levels" I thought it was mixing efficiency.*

**AR: This phrase has been changed to "mixing efficiency" throughout the manuscript.**

*RC P22L17: Be more specificâ˘A˘Te.g. " As both the residual circulation and the mixing are driven by wave breaking, a weaker residual circulation likely correlates with less mixing and thus longer mixing timescales"*

**AR: We agree with your suggested change.**

**Original sentence: "A reduced mean circulation would likely correspond to less mixing and longer mixing times."**

**New sentence: "As both the residual circulation and the mixing are driven by wave breaking, a weaker residual circulation likely correlates with less mixing and thus longer mixing timescales."**

*RC P22L25-6: The implication here and elsewhere in this section is that there is some knob to turn to "change" w\* or epsilon. There obviously isn't, and so while this paper has diagnosed that the mixing is too weak in JJA, for example, it hasn't come close to determining changes required in the CMAM30HR simulations to match the observations. Certainly changing the language here and P23L14, L29, P24L7 is necessary. If feasible, some discussion of what does set w\* (epsilon is probably harder) would be great. If anyone has looked at EP flux divergence esp. broken down by wavenumber, even in CMAM and not necessarily CMAM30, that would be a great thing to discuss here. The authors have done plenty to warrant publication and so they don't need to do it if it hasn't been done. Regardless, some discussion of what does drive the circulation and the mixing in the model would be appropriate here.*

**AR: While we cannot pinpoint an exact mechanism that would improve the CMAM30 simulations, the TLP model does provide a pathway or blueprint to make changes in CMAM that you wouldn't have otherwise. The end of Section 6 has been edited to provide more explanation of what could lead to improvements in CMAM.**

**Original sentences: "In particular, JJA appears to require increased mixing in all regions studied, implying that there is a substantial deficiency in the CMAM30HR simulation during this season. The most significant physical mechanism for mixing during this season is the Asian monsoon. The quality of this transport mechanism has not been directly assessed in CMAM30. It is unclear as to whether the mechanism has a direct or indirect effect but the Asian monsoon is a prominent climatological feature of the upper troposphere and lower stratosphere at this time of the year and it can be speculated that the required additional mixing may be related to the strength or extent of the simulated monsoon."**

**New sentences: "It follows that if *w\** needs to be reduced in the model then a reduction in wave activity is required.  There are specific waves that break in the lower, middle, and upper stratosphere that could be investigated for possible sources of increased *w\**.  For mixing changes, the background state of the winds and corresponding critical layers for wave breaking could be investigated for critical layers that extend too far into the tropics."**

*RC P24L23: If you've proven that the BDC is too rapid, say so here. As far as I recall, you said that it might be too strong.*

**AR: The sentence has been modified to more clearly state that the BDC is too rapid in CMAM30.**

**Original sentence: "The polar vortex comparisons reveal issues in both the timing and strength of the downwelling portion of the deep branch, which is likely directly related to the too-rapid overturning nature of CMAM's BDC."**

**New sentence: "The polar vortex comparisons reveal issues in both the timing and strength of the downwelling portion of the deep branch, which is related to the too-rapid BDC in CMAM30 simulations observed in the zonal mean comparisons."**

*RC P24L33-4: Nonsense. Insufficient mixing does not cause any changes to the BDC (at least to first order˘A˘ Tsecond order effects on ozone and the corresponding heating due to ozone might be minor but that has definitely not been addressed in this paper).*

**AR: We agree that this is not an accurate statement based on the analysis presented here. We have decided to review the entire paragraph to make it a more accurate summary of our findings.**

**Original paragraph: "Insufficient mixing during JJA may be related to the poorly simulated tropospheric intrusions during the same season and this same issue may be directly related to the younger air in CMAM30HR. Garny et al. (2014) found that, in the subtropical lower stratosphere, younger air is the result of a combination of a speeding up of the overturning circulation and weaker mixing or recirculation of the stratospheric air between the tropics and midlatitudes. This may be evidence for insufficient mixing in the specified dynamics simulations being the cause of a too-rapid BDC. It is important to scrutinize the mixing levels in CCMs and GCMs since it appears to be related to the mechanisms driving the projected trends in stratospheric circulation, thereby influencing the simulations of stratospheric ozone recovery and climate change. The techniques used in this work, including the advanced sampling and use of the tropical leaky pipe model, have proven illuminating. It is suggested that other CCMs and GCMs investigate the use of these techniques in future studies."**

**New paragraph: "The analysis presented here highlights the importance of scrutinizing the mixing efficiency in CCMs and GCMs since it may be related to the mechanisms driving the projected trends in stratospheric circulation, thereby influencing the simulations of stratospheric ozone recovery and climate change. The techniques used in this work, including the advanced sampling and use of the tropical leaky pipe model, have proven illuminating. It is suggested that other CCMs and GCMs investigate the use of these techniques in future studies."**

---

## Author Response (AR2)

Response to comments by Referee #2: Toohey, Matthew on "Assessing stratospheric transport in the CMAM30 simulations using ACE-FTS measurements" by Felicia Kolonjari et al.

*RC = Referee Comment*
**AR = Author Response**

*RC: I commend the authors for successfully addressing so many of the reviewers comments and suggestions.*

**AR: We thank Referee #2 for acknowledging our efforts. We think the paper has been substantially improved by the thoughtful feedback provided through the review process.**

*RC: One (relatively minor) issue still remains unfilled in my eyes, and it concerns the explanation of differences brought about by the three different sampling strategies presented in Sec. 3.3. According to text there:*
*- The 'basic' sampling method involved selecting a vertical column based on the nearest neighbour grid point to the 30 km tangent point location with no interpolation and no consideration of the vertical extent of the profile.*
*- The 'intermediate' level of sampling extracted the vertical column based on a bilinear interpolation of the four closest grid points to the 30 km geometric tangent point but with no consideration of the variation in geographical location of the tangent points above or below 30 km.*
*- The 'advanced' sampling method improves on the intermediate level of sampling by performing the bilinear interpolation at each level of the ACE-FTS profile using the distinct geographic locations derived from the refraction model for the respective level.*

*So, based on these descriptions, the differences in mean N2O zonal mean cross-sections (Fig 4) based on sampling can be understood as follows:*
*- Advanced-Intermediate represents the impact of including the variation of the geographical location of tangent point with height.*
*- Advanced-Basic represents the combined impact of including the variation of geographical location of tangent point with height (as in Advanced-Intermediate) plus the impact of simply interpolating to the location of the 30 km tangent point (in other words, not using the nearest-neighbour as done in Basic).*

*The impact of including the variation of the geographical location of tangent point with the height of the retrieved profile, shown by the Advanced-Intermediate plot, is zero at ~30 km, as per definition, and pretty small otherwise (~2%). The Advanced-Basic plot shows differences that are much larger (reaching ~10%), which, by elimination, must be mostly the result of performing the horizontal interpolation and not using the nearest-neighbour.*

*This is a long-winded explanation for why I find the statement at pg 14, l13-14, and the discussion following, to be incorrect, or at least misleading. The main reason for differences between the Advanced and Basic methods must be mostly due to the horizontal interpolation to the 30 km tangent point. The horizontal variation of tangent point with altitude plays a relatively smaller role. The discussion here should make that clear.*

**AR: We appreciate this explanation and agree that the main reason for the differences between the advanced and basic methods is the horizontal bilinear interpolation. The text has been changed as follows.**

**Original text: "Since the basic sampling is limited to the 30 km tangent altitude, the comparison to the advanced technique in Fig. 4b reflects the influence of the geographical extent of the ACE-FTS profiles."**

**New text: "The comparison to the advanced technique in Fig. 4b reflects the combined influence of the horizontal interpolation and, to a lesser degree, the geographical extent of the ACE-FTS profiles."**

*Minor corrections:*

*RC: pg15, l16: there doesn't appear to be a difference in the scale of Fig 7c.*

**AR: The text has been changed as follows.**

**Original text: "Each of the panels shows the differences as a percentage, but note that the scale for Fig. 7c is different. This is because the CFC-11 relative differences become large in the stratosphere where concentrations tend toward zero. "**

**New text: "Each of the panels shows the differences as a percentage."**

*RC: pg16, l19: "tropical upwelling occurs"*

**AR: The text has been changed as follows.**

**Original text: "These results support the understanding that the most rapid tropical upwelling is occurs in the summer hemisphere as first reported by Yulaeva et al. (1994)."**

**New text: "These results support the understanding that the most rapid tropical upwelling occurs in the summer hemisphere as first reported by Yulaeva et al. (1994)."**

*RC: pg 17, l24: most-> mostly ?*

**AR: The text has been changed as follows.**

**Original text: "Tracers with a stratospheric sink are most depleted from the deep branch because they have had the most time for chemical loss to occur since they entered the stratosphere."**

**New text: "Tracers with a stratospheric sink are mostly depleted from the deep branch because they have had the most time for chemical loss to occur since they entered the stratosphere."**

We have also made additional corrections (primarily typographical and formatting) and added an acknowledgement to the manuscript. These changes are noted in the difference file.

[revised manuscript text omitted]